# Axon guidance genes modulate neurotoxicity of ALS-associated UBQLN2

Sang Hwa Kim[1]*, Kye D Nichols[1], Eric N Anderson[2], Yining Liu[1], Nandini Ramesh[2], Weiyan Jia[1], Connor J Kuerbis[1], Mark Scalf[3], Lloyd M Smith[3], Udai Bhan Pandey[2], Randal S Tibbetts[1]*

[1]Department of Human Oncology, University of Wisconsin School of Medicine and Public Health, Madison, United States; [2]Department of Pediatrics, Children's Hospital of Pittsburgh, University of Pittsburgh Medical Center, Pittsburgh, United States; [3]Department of Chemistry, University of Wisconsin-Madison, Madison, United States

**Abstract** Mutations in the ubiquitin (Ub) chaperone *Ubiquilin 2 (UBQLN2)* cause X-linked forms of amyotrophic lateral sclerosis (ALS) and frontotemporal dementia (FTD) through unknown mechanisms. Here, we show that aggregation-prone, ALS-associated mutants of UBQLN2 (UBQLN2[ALS]) trigger heat stress-dependent neurodegeneration in *Drosophila*. A genetic modifier screen implicated endolysosomal and axon guidance genes, including the netrin receptor, Unc-5, as key modulators of UBQLN2 toxicity. Reduced gene dosage of *Unc-5* or its coreceptor *Dcc/frazzled* diminished neurodegenerative phenotypes, including motor dysfunction, neuromuscular junction defects, and shortened lifespan, in flies expressing UBQLN2[ALS] alleles. Induced pluripotent stem cells (iPSCs) harboring UBQLN2[ALS] knockin mutations exhibited lysosomal defects while inducible motor neurons (iMNs) expressing UBQLN2[ALS] alleles exhibited cytosolic UBQLN2 inclusions, reduced neurite complexity, and growth cone defects that were partially reversed by silencing of *UNC5B* and *DCC*. The combined findings suggest that altered growth cone dynamics are a conserved pathomechanism in UBQLN2-associated ALS/FTD.

*For correspondence: shkim9@wisc.edu (SHK); rstibbetts@wisc.edu (RST)

Competing interest: The authors declare that no competing interests exist.

## Editor's evaluation

This valuable study carried out a genetic screening of *Drosophila* lines expressing wild-type or ALS/FTD mutations of ubiquilin 2 and identified several suppressors and enhancers of ubiquilin 2 phenotypes. The study particularly focused on two genes involved in axon guidance pathways, unc5, and beat-1b. The evidence supporting the conclusions is solid and will be of interest to a broad audience studying ALS/FTD and neurodegenerative diseases.

## Introduction

### Ubiquilins and proteostasis

Defective protein folding and proteostatic stress are common pathogenic mechanisms linking genetically and anatomically diverse neurodegenerative diseases (*Ling et al., 2013*). The steady state levels—and ultimately neurotoxicity—of aggregation-prone proteins are determined through a balance of protein production and protein clearance (*Prahlad and Morimoto, 2009*). The convergence of aging-dependent declines in protein degradation and nuclear import with environmental stresses may push this equation toward irreversible protein aggregation that sets the stage for neurodegenerative processes (*Morimoto, 2008*). Thus, enhancing protein degradation capacity—or reducing the aggregation potential of aggregation-prone proteins—represents a promising therapeutic avenue for amyotrophic lateral sclerosis (ALS) and other neurodegenerative proteinopathies.

The highly conserved ubiquilin (UBQLN) gene family fulfills diverse roles in protein folding, shuttling, and degradation (*Zheng et al., 2020*; *Rothenberg and Monteiro, 2010*). All eukaryotic UBQLNs feature an amino-terminal ubiquitin-like (UBL) domain and a carboxyl-terminal ubiquitin-associated (UBA) domain separated by a low complexity, methionine-rich central region harboring variable numbers of STI1-like repeats first identified in the yeast stress-inducible 1 (Sti1) protein and later described in several distinct classes of protein chaperones (*Fry et al., 2021*; *Zientara-Rytter and Subramani, 2019*; *Howe et al., 2020*; *Schmid et al., 2012*; *Li et al., 2013*). Mammals encode four ubiquilin proteins of which three (UBQLN1, UBQLN2, and UBQLN4) are widely expressed. UBQLN1 and UBQLN2 share >70% amino acid identity and are presumed to function in semi-redundant fashion, whereas UBQLN4 is a more distantly related paralog.

Although specific functions of individual ubiquilins in mammals are largely unknown, a generic model for UBQLN function holds that the UBA domain engages ubiquitylated substrate while the UBL domain engages the proteasome, leading to substrate degradation (*Itakura et al., 2016*). STI1-like repeats form a hydrophobic groove that is thought to engage hydrophobic regions of client proteins (*Fry et al., 2021*). It has been reported that UBQLN deficiency (Df) leads to defects in autophagy and ER-associated protein degradation (*Rothenberg et al., 2010*; *Lim et al., 2009*; *N'Diaye et al., 2009*; *Lee et al., 2013*). The central methionine-rich domain has been shown to bind transmembrane domains of mitochondrial proteins, which appears central to their UBQLN-dependent triage and degradation (*Itakura et al., 2016*).

## UBQLN2 mutations in ALS/dementia

Interest in ubiquilin function was greatly stimulated by the discovery that dominant mutations in UBQLN2 cause X-linked ALS/frontotemporal dementia (FTD) (*Deng et al., 2011*; *Gellera et al., 2013*). Most ALS-associated mutations in UBQLN2 are clustered within 42-amino acid proline-rich repeat (PRR) that is unique to UBQLN2 (*Deng et al., 2011*) however, disease-linked mutations outside this region have also been described (*Synofzik et al., 2012*; *Daoud et al., 2012*). In addition, UBQLN2-mutant patients exhibit a range of phenotypes that includes FTD, ALS, and spastic paraplegia (*Gkazi et al., 2019*). Interestingly, ubiquilin-positive aggregates are a near universal occurrence in TDP-43-positive ALS, as well as ALS linked to *C9ORF72* expansions (C9-ALS) (*Brettschneider et al., 2012*). These correlative findings suggest that ubiquilin pathology may contribute to the molecular pathogenesis of ALS even in the absence of *UBQLN2* gene mutations.

Transgenic or virus-directed expression of UBQLN2^ALS mutants in rodents recapitulates protein UBQLN2 inclusions seen in ALS/FTD patients and elicits variable phenotypic abnormalities ranging to mild gait and memory defects to neuronal loss, paralysis, and early death (*Gorrie et al., 2014*; *Wu et al., 2015*; *Huang et al., 2016*; *Le et al., 2016*; *Sharkey et al., 2020*). Coexpression of UBQLN2^P497H and an ALS-associated TDP-43 allele under control of the neurofilament heavy gene promoter caused severe motor neuron loss and muscle wasting (*Picher-Martel et al., 2019*). Brain-directed expression of wild-type UBQLN2 also caused toxicity phenotypes (*Huang et al., 2016*; *Le et al., 2016*; *Sharkey et al., 2020*), potentially due to disruptions in Ub homeostasis. Finally, like many proteins implicated in ALS/FTD (*Pakravan et al., 2021*), UBQLN2 harbors low complexity regions and undergoes liquid-liquid phase separation (*Sharkey et al., 2018*; *Dao et al., 2018*). While ALS-associated mutations in the PRR may interfere with UBQLN2 liquid demixing, the relevance to disease pathogenesis is presently unclear.

In previous work, we reported that expression of ALS-associated *UBQLN2* mutants caused mutation-dependent neurotoxicity in *Drosophila* (*Kim et al., 2018*). Here, we carried out genetic screens for UBQLN2 toxicity modifiers using flies expressing UBQLN2^ALS alleles with differing aggregation potential. Suppressor genes emerging from this screen were then tested for impacts on the toxicity of endogenous UBQLN2^ALS mutants in iMNs. Our findings suggest that endolysosomal dysfunction and axon guidance defects are phenotypic drivers of neurodegeneration in UBQLN2-associated ALS/FTD.

## Results

### An aggregation-prone UBQLN2$^{4XALS}$ allele exhibits heat stress-dependent eye toxicity

We previously reported that homozygous expression of single-copy human UBQLN2$^{ALS}$ alleles caused mild eye toxicity when expressed under control of an eye-specific GMR driver at 22°C (*Kim et al., 2018*). Reasoning that UBQLN2-associated phenotypes may be worsened by heat stress (HS), we compared eye morphologies between flies expressing UBQLN2$^{WT}$, a clinical UBQLN2$^{P497H}$ allele, and a highly aggregation-prone UBQLN2$^{4XALS}$ mutant that harbors four different clinical mutations (P497H, P506T, P509S, and P525S, *Figure 1A*; *Kim et al., 2018*). None of the UBQLN2 transgenes caused an overt external eye phenotype when expressed from a hemizygous GMR>UBQLN2 Chr2 locus at 22°C (*Figure 1B*). By contrast, GMR>UBQLN2$^{4XALS}$ flies—but not GMR>UBQLN2$^{WT}$ or GMR>UBQLN2$^{P497H}$ flies—exhibited a moderately severe rough eye (RE) phenotype that was characterized by eye depigmentation and loss of ommatidial facets in both male and female flies (*Figure 1B*). As expected, UBQLN2$^{4XALS}$ was significantly less soluble than UBQLN2$^{WT}$ and UBQLN2$^{P497H}$ at both 22°C and 29°C; however, the relative proportion of insoluble UBQLN2$^{4XALS}$ was comparable at both temperatures, suggesting that increased aggregation of UBQLN2$^{4XALS}$ cannot wholly account for its enhanced toxicity at 29°C (*Figure 1—figure supplement 1A, B*).

We also performed quantitative mass spectrometry (MS) to assess relative abundance of UBQLN2$^{WT}$, UBQLN2$^{4XALS}$, and endogenous *Drosophila* Ubqln (dUbqln) in whole-head extracts. The average number of peptide spectral matches for hUBQLN2 was comparable between GMR>UBQLN2$^{WT}$ and GMR>UBQLN2$^{4XALS}$ flies reared at 22°C and 29°C (*Figure 1—figure supplement 1C*). hUBQLN2-unique peptides were ~6-fold more abundant than dUbqln peptides in both GMR>UBQLN2$^{WT}$ and GMR>UBQLN2$^{4XALS}$ heads, providing a lower limit of hUBQLN2 overexpression (*Figure 1—figure supplement 1C*). As expected, given the heat inducibility of UAS, absolute numbers of hUBQLN2 peptides were higher in GMR>UBQLN2$^{WT}$ and GMR>UBQLN2$^{4XALS}$ flies reared at 29°C versus 22°C; however, the difference was not statistically significant across three replicates. Altogether these findings established HS-dependent toxicity of UBQLN2$^{4XALS}$ that is not solely due to increased protein expression.

### Transcriptomic analysis of UBQLN2$^{ALS}$ flies

We next performed RNA-Seq analysis of whole heads from GMR>UBQLN2$^{WT}$, GMR>UBQLN2$^{P497H}$, GMR>UBQLN2$^{4XALS}$. Each of the three GMR>UBQLN2 lines exhibited hundreds of gene expression changes relative to GMR-Gal4 controls, suggesting that UBQLN2 overexpression has a substantial impact on cellular regulation (*Figure 1—figure supplement 2A*). Consistent with their more severe eye phenotype, GMR>UBQLN2$^{4XALS}$ flies exhibited a distinct RNA-Seq gene expression signature relative to GMR>UBQLN2$^{WT}$ and GMR>UBQLN2$^{P497H}$ flies, which clustered together in principle component analysis (*Figure 1—figure supplement 2B*). Using an FDR of <0.05, 629 genes were uniquely changed in GMR>UBQLN2$^{4XALS}$ flies versus GMR>UBQLN2$^{WT}$, GMR>UBQLN2$^{P497H}$, and GMR-Gal4 controls at 29°C (*Figure 1—figure supplement 2C*, *Figure 1—figure supplement 2—source data 1*). Of these, 402 genes were upregulated and 227 genes were downregulated in GMR>UBQLN2$^{4XALS}$ flies (*Figure 1—figure supplement 2D, E*, Supplementary Dataset 1 and 2, respectively). Gene ontology (GO) analysis revealed that mRNAs involved in eye development and phototransduction, including inaC, ninaE, and Rh3, were broadly downregulated in GMR>UBQLN2$^{4XALS}$ flies, likely reflecting degenerative cell loss (*Figure 1—figure supplement 2D, E*, Supplementary Dataset 2). Upregulated GO terms included response to biotic stimulus, which includes *Drosophila* innate immunity genes such as *Dro*, *AttB*, *AttC*, and *Listericin*, and response to UV light, which includes the small heat-shock protein-encoding genes *Hsp23*, *Hsp26*, and *Hsp27* (*Figure 1—figure supplement 2D, E*, Supplementary Dataset 1). Upregulation of HSPs may be driven by UBQLN2$^{4XALS}$ misfolding. In contrast to the profound differences between GMR>UBQLN2$^{4XALS}$ and GMR>UBQLN2$^{WT}$ flies, only 64 genes were differentially expressed between GMR>UBQLN2$^{WT}$ and GMR>UBQLN2$^{P497H}$ flies, which is consistent with their qualitatively similar eye phenotypes. Finally, 28 differentially expressed genes were common to GMR>UBQLN2$^{4XALS}$ and GMR>UBQLN2$^{P497H}$ flies relative to GMR>UBQLN2$^{WT}$ flies (*Figure 1—figure supplement 2F*). Overall, these findings reveal that the strong eye phenotype of GMR>UBQLN2$^{4XALS}$ is accompanied by robust changes in gene expression.

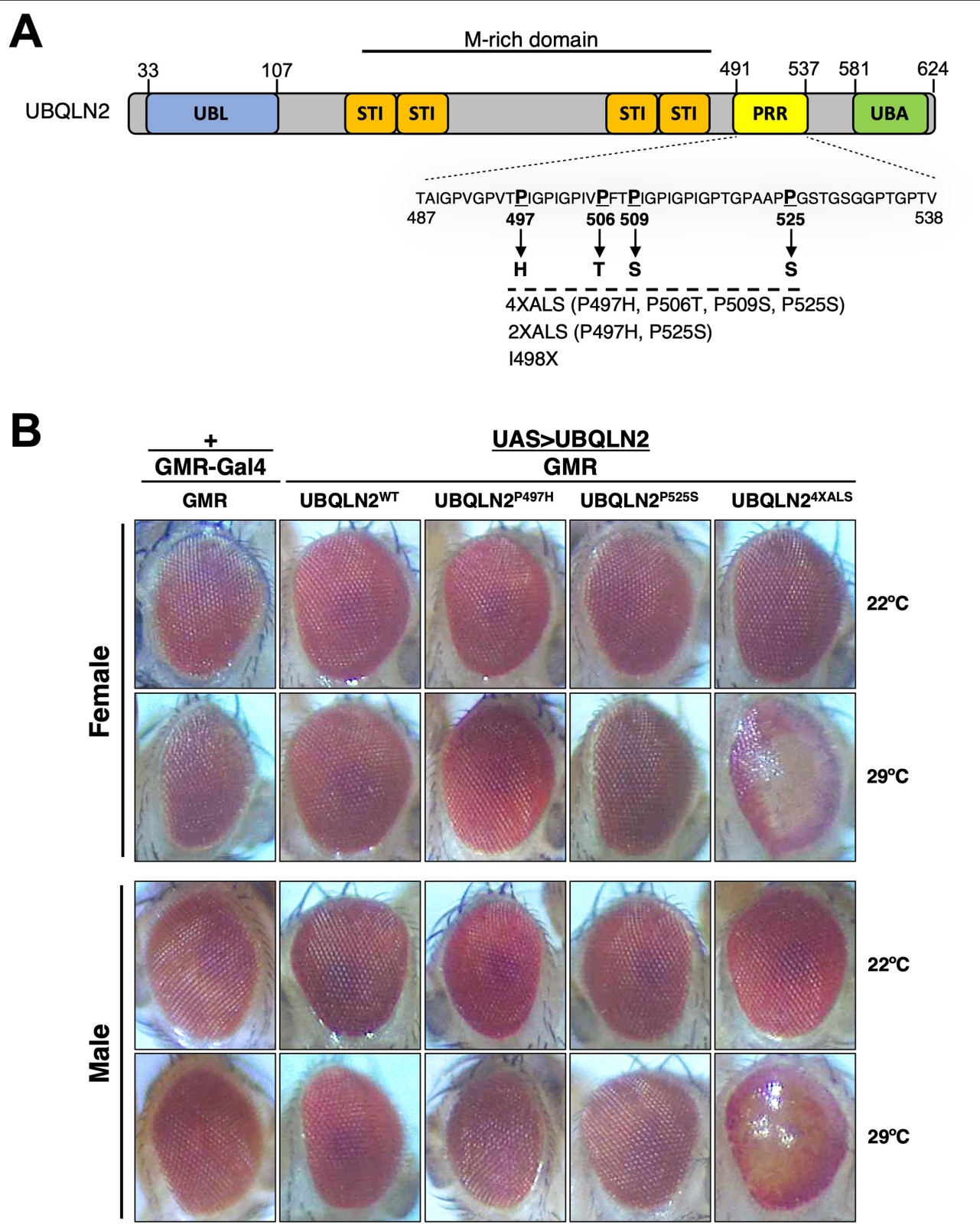

**Figure 1.** UBQLN2 exerts heat stress (HS)-dependent toxicity. (**A**) Schematic of UBQLN2 and amyotrophic lateral sclerosis (ALS)-associated mutations. Approximate locations of ubiquitin-like (UBL); STI1-like (STI); proline-rich repeat (PRR); and ubiquitin-associated (UBA) domains are shown, as are ALS-associated mutations investigated in this study. (**B**) Eye images from flies expressing UBQLN2[WT], UBQLN2[P497H], UBQLN2[P525S], or UBQLN2[4XALS] under control of the eye-specific GMR driver at 22°C and 29°C. Note depigmentation and destruction of ommatidial facets in UBQLN2[4XALS] flies reared at 29°C.

*Figure 1 continued*

The online version of this article includes the following source data and figure supplement(s) for figure 1:

**Figure supplement 1.** Ubiquitin (Ub)-binding contributes to eye degeneration and aggregation of UBQLN2$^{ALS}$ mutants.

**Figure supplement 1—source data 1.** Uncropped Western blot images corresponding to *Figure 1—figure supplement 1A*, and PSM corresponding to *Figure 1—figure supplement 1C*.

**Figure supplement 2.** Gene expression profiles of GMR>UBQLN2$^{ALS}$ flies.

**Figure supplement 2—source data 1.** Gene expression data sets corresponding to *Figure 1—figure supplement 2*.

## Df screens for UBQLN2$^{ALS}$ modifier genes

The RE phenotype of GMR>UBQLN2$^{4XALS}$ flies allowed us to perform genetic modifier screens. To this end, we screened a Bloomington Df library of 194 lines spanning 85% of chromosome 2 (*Figure 2A*). The screen was carried out at 29°C and eye phenotypes were scored on a scale of 1–5, with a score of '1' representing a morphologically normal eye; a score of '3' representing the unmodified UBQLN2$^{4XALS}$ eye phenotype; and a score of '5' representing eyes harboring >50% necrotic patches (*Figure 2B*). Lethal enhancers were also noted. While the mild eye phenotype of GMR>UBQLN2$^{P497H}$ flies largely precluded identification of suppressors, the side-by-side screening of this line allowed us to identify shared and/or mutation-specific enhancers. Candidate modifier Dfs were subjected to secondary screens against UBQLN2$^{WT}$, UBQLN2$^{P497H}$, and UBQLN2$^{4XALS}$.

The UBQLN2$^{4XALS}$ screen identified seven suppressors, three of which overlapped a common genomic interval, and 23 enhancers, including 15 lethal enhancers (*Figure 2C*). The UBQLN2$^{P497H}$ screen identified six enhancers, two of which were also identified in the UBQLN2$^{4XALS}$ screen (*Figure 2D*). Notably, all Dfs that enhanced the GMR>UBQLN2$^{P497H}$ eye phenotype also caused enhanced eye phenotypes in GMR>UBQLN2$^{WT}$ flies.

A combination of iterative Df screening and RNAi screening was then used to map causal genes within three UBQLN2$^{4XALS}$ suppressor loci (BSC180, ED2426, and Exel6038) and one enhancer locus common to UBQLN2$^{4XALS}$ and UBQLN2$^{P497H}$ (BSC37) (*Figure 2C and D*). Among 17 annotated genes within BSC37 (*Figure 2—figure supplement 1A*), we prioritized *Rab5*, which encodes an early endosomal protein that interacts with the ALS2 gene product, Alsin (*Hsu et al., 2018*; *Kunita et al., 2007*). In support of a role for *Rab5* as a UBQLN2 modifier gene, *Rab5* knockdown phenocopied the hyperpigmented phenotype seen in GMR>UBQLN2$^{4XALS}$ flies crossed to BSC37 at 29°C (*Figure 2—figure supplement 1B, C*). *Rab5* knockdown also caused a hyperpigmented eye phenotype in GMR>UBQLN2$^{WT}$ and GMR>UBQLN2$^{P497H}$ flies (*Figure 2—figure supplement 1C*), indicating *Rab5* is a mutation-independent UBQLN2 enhancer. Interestingly, while *Rab5* knockdown also caused hyperpigmented eye patches in GMR>UBQLN2$^{WT}$, and GMR>UBQLN2$^{P497H}$ flies reared at 22°C, GMR>UBQLN2$^{4XALS}$/shRab5 flies reared at 22°C failed to exhibit eye patches (*Figure 2—figure supplement 1C*). This finding supports the idea that UBQLN2$^{4XALS}$ phenotypes are HS dependent and is consistent with an earlier study showing stronger eye phenotypes in GMR>UBQLN2$^{WT}$ and GMR>UBQLN2$^{P497H}$ flies relative to GMR>UBQLN2$^{4XALS}$ flies at 22°C (*Kim et al., 2018*). Finally, we showed that overexpression of GFP-Rab5 partially rescued the eye phenotype of GMR>UBQLN2$^{WT}$, GMR>UBQLN2$^{P497H}$, and GMR>UBQLN2$^{4XALS}$ flies reared at 29°C (*Figure 2—figure supplement 1F*). Rab5 knockdown or overexpression did not affect UBQLN2 expression (*Figure 2—figure supplement 1D, E, G, H*). The combined findings suggest that *Rab5* is a general modulator of UBQLN2 toxicity and that UBQLN2 overexpression interferes with endolysosomal function in flies.

A similar approach was used to map the causal UBQLN2$^{4XALS}$ suppressor gene in BSC180. Two different overlapping Dfs (C144 and ED4651) rescued UBQLN2$^{4XALS}$ eye toxicity to a similar extent as BSC180 (*Figure 2—figure supplement 2A, B*). Within this overlap we identified *lilliputian* (*lilli*) as a gene of interest. *lilli* encodes a transcriptional elongation factor that is orthologous to mammalian *AF4/FMR2* (fragile X mental retardation 2) (*Wittwer et al., 2001*). Interestingly, *lilli* mutations were recently shown to suppress toxicity of TDP-43 and *C9ORF72*-derived dipeptide repeat proteins (DPRs) in *Drosophila* (*Chung et al., 2018*; *Yuva-Aydemir et al., 2019*). In support of *lilli* as a UBQLN2$^{4XALS}$ suppressor, a *lilli*$^{A17-2}$ LOF allele diminished UBQLN2$^{4XALS}$-mediated eye toxicity to a similar extent as *lilli*-spanning Dfs (*Figure 2—figure supplement 2B, C*). Thus, *lilli* may be of interest as a general suppressor of toxicity arising from misexpression of neurodegeneration-associated genes in *Drosophila*.

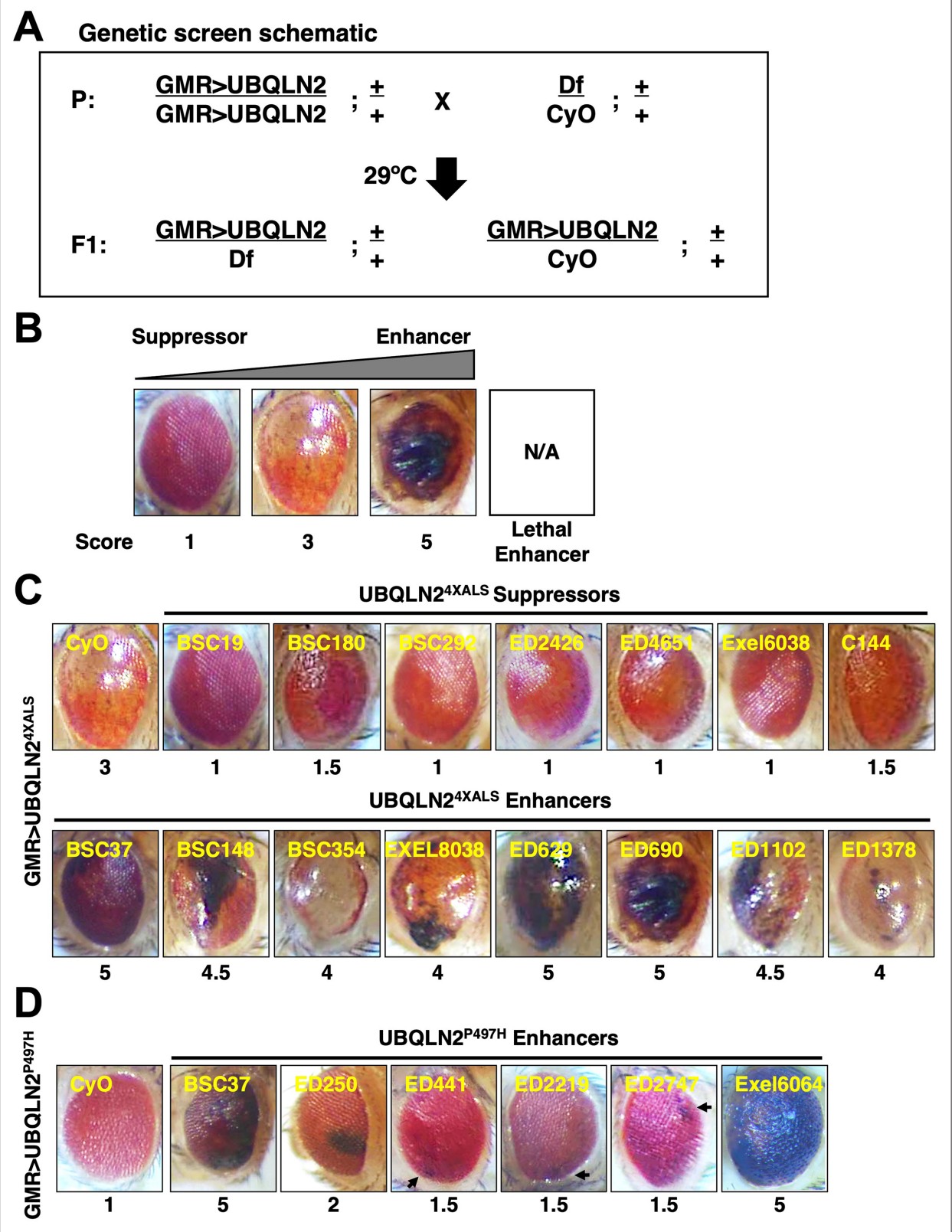

**Figure 2.** Identification of UBQLN2 modifier genes. (**A**) Schematic of deficiency (Df) screen for UBQLN2 modifiers. (**B**) Representative eye images and scoring rubric for F1 progeny of GMR>UBQLN2$^{4XALS}$ flies crossed to Df lines. (**C**) Representative UBQLN2$^{4XALS}$ modifier genes. Eye images were taken of F1 progeny from crosses of GMR>UBQLN2$^{4XALS}$ to indicated Df lines at 1–3 days post eclosion. Suppressors and enhancers are shown in top and bottom

*Figure 2 continued on next page*

*Figure 2 continued*

rows, respectively. (**D**) Representative eye images of UBQLN2$^{P497H}$ enhancers. Arrows indicate foci of eye degeneration. Eye degeneration scores are displayed below each eye image (**B, C, D**).

The online version of this article includes the following source data and figure supplement(s) for figure 2:

**Source data 1.** List of Dfs that modified GMR>UBQLN2$^{P497H}$ and/or GMR>UBQLN2$^{4XALS}$ eye phenotypes corresponding to *Figure 2C, D*.

**Figure supplement 1.** *Rab5* is a UBQLN2$^{4XALS}$ modifier gene.

**Figure supplement 1—source data 1.** Uncropped Western blot images corresponding to *Figure 2—figure supplement 1D, E, G, H*.

**Figure supplement 2.** Transcriptional elongation factor lilliputian (*lilli*) is a UBQLN2$^{4XALS}$ suppressor.

**Figure supplement 2—source data 1.** Uncropped Western blot images corresponding to *Figure 2—figure supplement 2C*.

### *Unc-5* mutations suppress UBQLN2 toxicity

Next, we mapped a UBQLN2$^{4XALS}$ suppressor in the genetic interval spanned by ED2426 and BSC346, which suppressed GMR>UBQLN2$^{4XALS}$ eye phenotypes to a similar extent (*Figure 3A and B*). The region of overlap between ED2426 and BSC346 contains *Unc-5*, which encodes a transmembrane dependence receptor that mediates axonal repulsion and apoptosis suppression in response to secreted netrin ligands (*Boyer and Gupton, 2018*; *Labrador et al., 2005*; *Keleman and Dickson, 2001*; *Llambi et al., 2005*; *Wang et al., 2009*; *Ahn et al., 2020*).

In support of a genetic interaction between *Unc-5* and UBQLN2$^{4XALS}$, two *Unc-5* LOF alleles (*Labrador et al., 2005*), and three different RNAi lines diminished the RE phenotype of GMR>UBQLN2$^{4XALS}$ flies without affecting UBQLN2 expression (*Figure 3C, D and E*). By contrast, *Unc-5* knockdown had no effect on the RE phenotype caused by the expression of ALS-associated FUS alleles that cause severe degenerative phenotypes in *Drosophila* (*Figure 3—figure supplement 1*; *Lanson et al., 2011*; *Kwiatkowski et al., 2009*; *Vance et al., 2009*). This suggests that genetic interaction with *Unc-5* may be specific to UBQLN2 versus other ectopically expressed ALS-associated proteins in *Drosophila*.

The repulsive activity of *Unc-5* on axon guidance is antagonized by *frazzled (fra),* which encodes the fly ortholog of mammalian *deleted in colon carcinoma* (*DCC*) (*Chan et al., 1996*; *Keino-Masu et al., 1996*; *Kolodziej et al., 1996*). *fra* silencing by two different RNAi lines rescued the UBQLN2$^{4XALS}$ eye phenotype to a similar extent as *Unc-5* knockdown (*Figure 3F*). The combined findings suggest that signaling through Unc-5/Frazzled potentiates UBQLN2$^{4XALS}$ toxicity in the *Drosophila* compound eye.

To evaluate the impact of *Unc-5* silencing on motor function, we measured climbing behavior of recombinant flies expressing UBQLN2$^{4XALS}$ under control of a D42 motor neuron driver in the presence of shUnc-5 or control shRNAs at 29°C. The moderate climbing defect of D42>UBQLN2$^{4XALS}$ flies relative to D42-Gal4 flies was partially reversed by an shUnc-5 (TRiP) allele in both male and female flies, while a second *Unc-5* shRNA line (KK) rescued climbing in male but not female flies (*Figure 4A*).

Consistent with their climbing defects, D42>UBQLN2$^{4XALS}$ flies exhibited neuromuscular junction (NMJ) abnormalities, including increased numbers of satellite boutons and reduced numbers of mature boutons. Both phenotypes were corrected by *Unc-5* silencing (*Figure 4B and C*). Finally, we also measured the effect of *Unc-5* silencing on lifespans of flies expressing UBQLN2$^{4XALS}$ under control of a pan-neuronal Elav driver. Elav>UBQLN2$^{4XALS}$/shUnc-5 flies showed a significant increase in lifespan relative to Elav>UBQLN2$^{4XALS}$ flies crossed to shLuci, with the effect being most pronounced in female flies (*Figure 4D*). *Unc-5* knockdown also modestly increased the lifespan of Elav-Gal4 male flies relative to Elav-Gal4/shLuci flies while having no impact on lifespan of Elav-Gal4 female flies (*Figure 4D*). Altogether, these experiments suggest that aberrant Unc-5 signaling contributes to neuronal phenotypes in UBQLN2$^{4XALS}$ flies.

### The motor neuron guidance factor *beat-1b* suppresses UBQLN2$^{4XALS}$ eye toxicity

The identification of *Unc-5* as a UBQLN2$^{4XALS}$ suppressor raised the possibility that axonal guidance defects are particularly relevant to the UBQLN2 toxicity mechanism. Interestingly, the overlapping genetic interval spanned by the UBQLN2$^{4XALS}$ suppressors Exel6038 and r10 contains *beat-1b* and *beat-1c*, two members of the *beaten path* (*beat*) family of axon guidance genes (*Figure 5A–C*; *Vactor et al., 1993*). Neuronally expressed Beat proteins regulate motor axon guidance and defasciculation

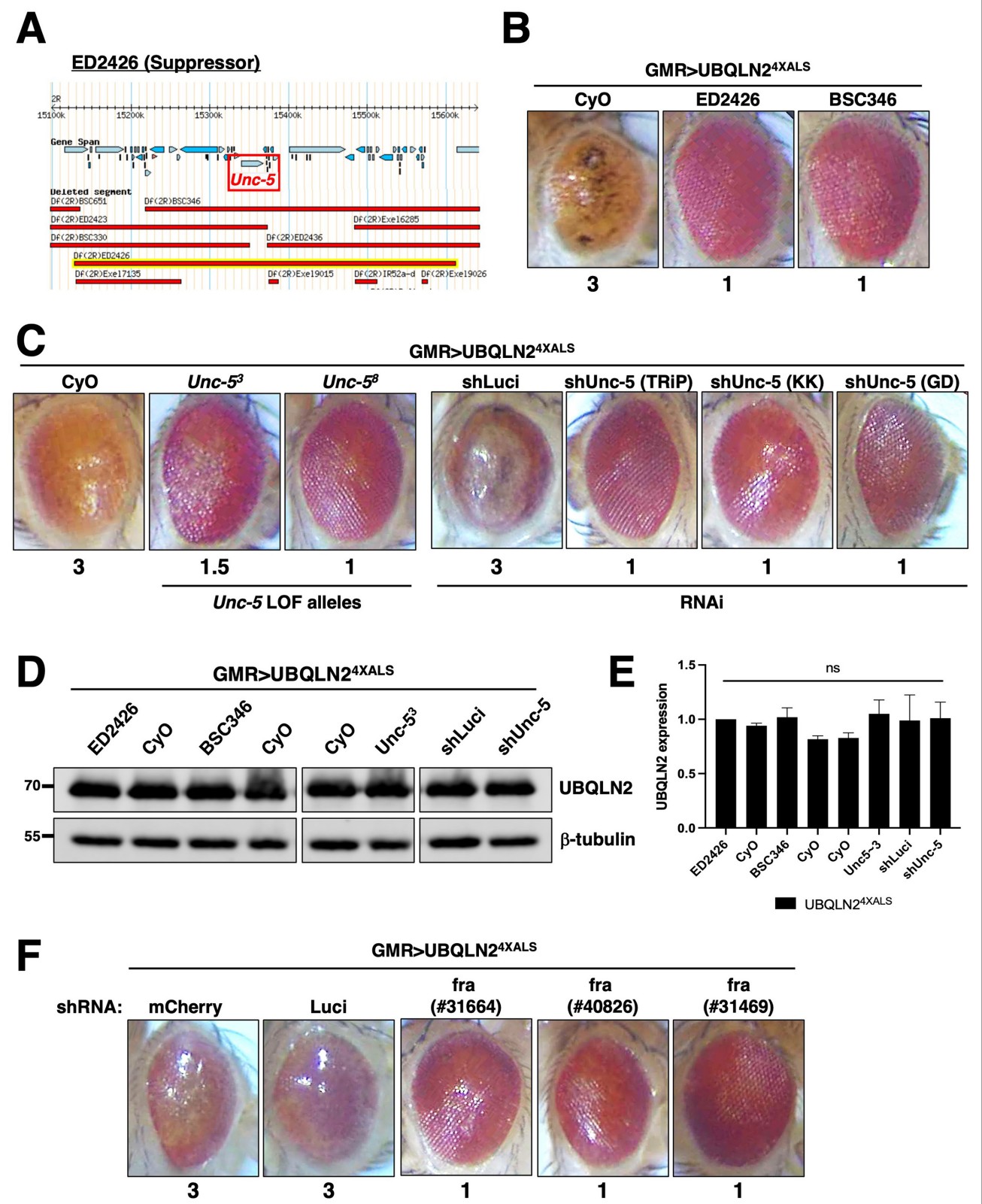

**Figure 3.** Disruption of *Unc-5* suppresses UBQLN2-associated eye degeneration. (**A**) Schematic depiction of ED2426 and overlapping deficiencies (Dfs) in relation to the *Unc-5* gene locus. (**B**) Representative eye images of GMR>UBQLN2⁴ˣᴬᴸˢ flies crossed to Df lines ED2426 and BSC346. (**C**) Single allele expression of *Unc-5* LOF alleles (left panels) or three independent *Unc-5* RNAi alleles diminished the rough eye (RE) phenotype of GMR>UBQLN2⁴ˣᴬᴸˢ flies at 29°C. (**D, E**) UBQLN2 expression levels in whole heads of GMR>UBQLN2⁴ˣᴬᴸˢ flies on the indicated genetic backgrounds. (**E**) Quantification of

*Figure 3 continued on next page*

eLife Research article

Cell Biology | Neuroscience

UBQLN2 expression normalized to β-tubulin. The bars represent mean with *SEM* of triplicate samples. Unpaired t-test was used for statistical analysis. (**F**) *fra* silencing reduced the RE phenotype of GMR>UBQLN2$^{4XALS}$ flies. Shown are representative eye phenotypes of F1 progeny from GMR>UBQLN2$^{4XALS}$ flies crossed to control (shLuci, shmCherry), or *fra* RNAi lines at 29°C. Eye degeneration scores are displayed below each eye image (**B, C, F**).

The online version of this article includes the following source data and figure supplement(s) for figure 3:

**Source data 1.** Uncropped Western blot images corresponding to *Figure 3D, E*.

**Figure supplement 1.** *Unc-5* silencing does not modify FUS-associated eye degeneration.

in response to Sidestep (Side) ligands expressed on target substrates (*Fambrough and Goodman, 1996*; *Sink et al., 2001*; *Siebert et al., 2009*; *de Jong et al., 2005*). UBQLN2$^{4XALS}$ flies crossed to a Beat-1b shRNA line exhibited less severe eye degeneration than UBQLN2$^{4XALS}$ flies expressing shRNAs targeting luciferase, mCherry, or *beat-1c* (*Figure 5D, E and F*). GMR>UBQLN2$^{4XALS}$ flies heterozygous for a p-element insertion in the *beat-1b* ORF also exhibited improved eye phenotype relative to control GMR>UBQLN2$^{4XALS}$ controls (*Figure 5D*). These findings support the idea that Beat-1b signaling contributes to UBQLN2 toxicity and further implicate axonal guidance defects as a contributing pathomechanism to UBQLN2-associated neurodegeneration.

## iPSC models for UBQLN2-associated ALS

We employed CRISPR/CAS9 to introduce ALS-associated mutations into the X-linked *UBQLN2* gene in neonatal male induced pluripotent stem cells (iPSCs) (WC031i-5907-6, see Materials and methods, and *Du et al., 2015*). During the course of this work, we serendipitously derived an *UBQLN2* allele harboring clinical P497H and P525S mutations (termed UBQLN2$^{2XALS}$) as well as a UBQLN2$^{I498X}$ allele that truncated the UBQLN2 ORF at codon 498 within the PRR. The expression and RIPA solubility of UBQLN2$^{P497H}$, UBQLN2$^{2XALS}$, and UBQLN2$^{4XALS}$ were comparable to UBQLN2$^{WT}$ in undifferentiated iPSCs, while UBQLN2$^{I498X}$ could not be detected by Western blotting suggesting it is a null allele (*Figure 6A*). We also failed to detect cytologic UBQLN2$^{P497H}$, or UBQLN2$^{4XALS}$ aggregates in immunostaining experiments, indicating that endogenous ALS mutations are insufficient to promote UBQLN2 aggregation in undifferentiated iPSCs (*Figure 6B*). As expected, UBQLN2$^{I498X}$ iPSCs exhibited very weak immunoreactivity with UBQLN2 antibodies.

Previous work demonstrated that the 4XALS mutation increased the chymotrypsin sensitivity of purified UBQLN2, likely due to defective folding of the PRR (*Kim et al., 2018*). To determine whether the 4XALS mutation altered the folding of endogenous UBQLN2, we incubated detergent extracts from UBQLN2$^{WT}$, UBQLN2$^{P497H}$, UBQLN2$^{4XALS}$ iPSCs with increasing concentrations of chymotrypsin. As shown in *Figure 6C*, UBQLN2$^{4XALS}$ exhibited a unique chymotryptic fragmentation pattern relative to UBQLN2$^{WT}$ and UBQLN2$^{P497H}$, suggesting that endogenous UBQLN2$^{4XALS}$ is misfolded.

Reasoning that transient inhibition of protein degradation may potentiate UBQLN2 aggregation, we prepared soluble and insoluble fractions of UBQLN2$^{WT}$, UBQLN2$^{P497H}$, and UBQLN2$^{4XALS}$ iPSCs exposed to the proteasome inhibitor MG132 or the autophagy inhibitor bafilomycin A1 (BafA1). While MG132 had no effect (not shown), BafA1 decreased the solubility of both wild-type and mutant UBQLN2 proteins, with the effect being most pronounced for UBQLN2$^{4XALS}$ (*Figure 6D*). In addition, while the solubility of UBQLN2$^{WT}$ and UBQLN2$^{P497H}$ recovered following BafA1 washout, the fraction of insoluble UBQLN2$^{4XALS}$ remained elevated (*Figure 6D and E*). Finally, the relative abundance of lipidated and non-lipidated forms of LC3 were comparable between UBQLN2$^{WT}$, UBQLN2$^{P497H}$, and UBQLN2$^{4XALS}$, and UBQLN2$^{I498X}$ iPSCs, suggesting similar levels of autophagic flux (*Figure 6D*).

We next evaluated the localization of wild-type and ALS-mutant UBQLN2 proteins to endolysosomal structures. UBQLN2$^{WT}$, UBQLN2$^{P497H}$, and UBQLN2$^{4XALS}$ partially localized with the lysosomal marker LAMP1 in BafA1-treated iPSCs (*Figure 6—figure supplement 1A*). By contrast, UBQLN2 punctae did not significantly overlap with Rab5-positive endosomes (*Figure 6—figure supplement 1B*). Given colocalization of UBQLN2 with LAMP1, we evaluated impacts of P497H and 4XALS mutations on lysosomal number and size in live cells using the lysosome-tropic fluorescent probe, LysoTracker. Both the abundance and average size of lysosomes were significantly elevated in UBQLN2$^{P497H}$ and UBQLN2$^{4XALS}$ iPSCs, relative to UBQLN2$^{WT}$ or UBQLN2$^{I498X}$ iPSCs, which exhibited qualitatively similar LysoTracker staining patterns (*Figure 6—figure supplement 1C*). In particular, the frequency of lysosomes greater than 5 μm and 10 μm was significantly elevated in UBQLN2$^{4XALS}$ iPSCs versus UBQLN2$^{P497H}$ iPSCs

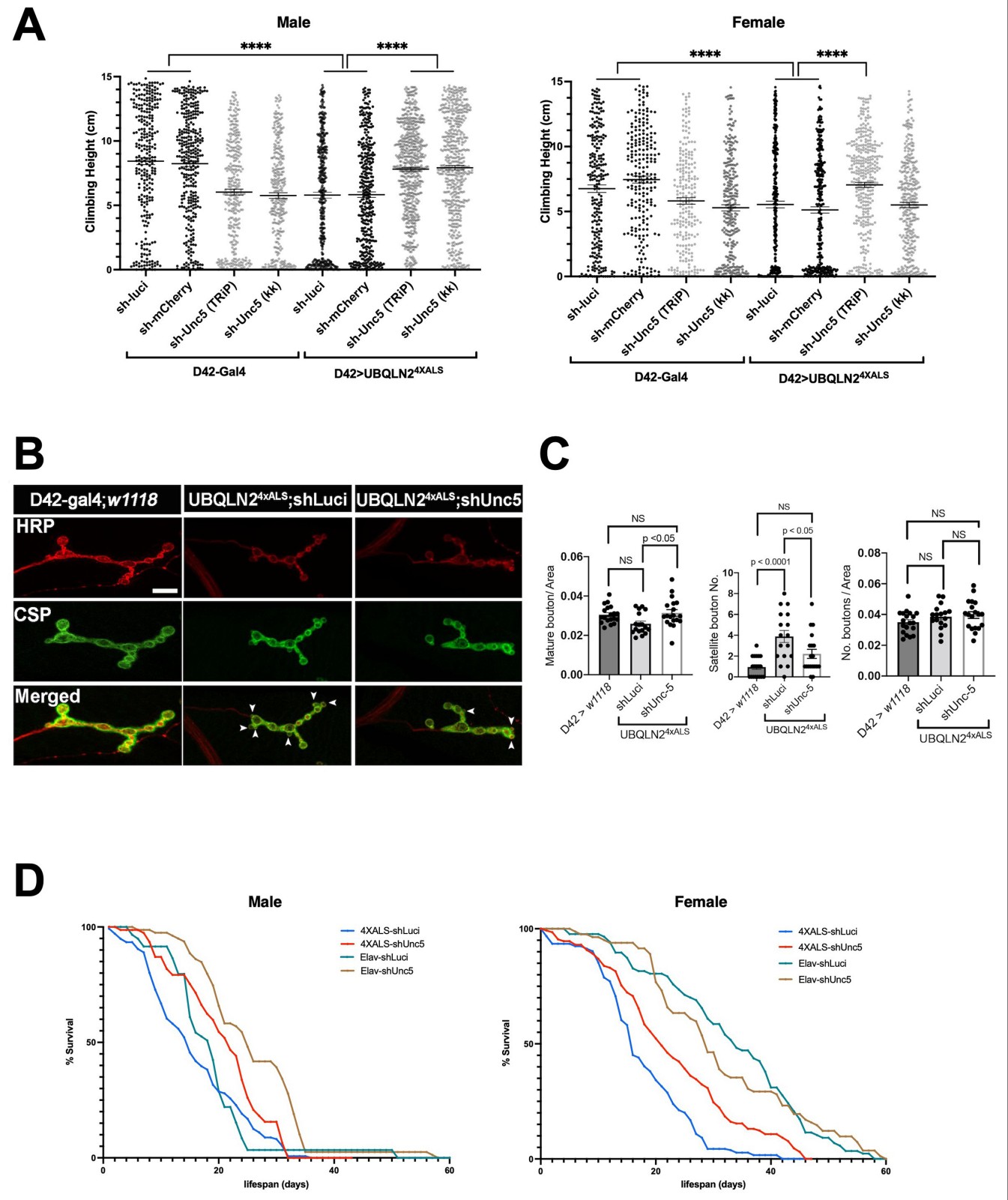

**Figure 4.** *Unc-5* silencing in *Drosophila* suppressed UBQLN2[4XALS]-associated neuronal phenotypes. (**A**) Recombinant D42>UBQLN2[4XALS] flies expressing UBQLN2 under control of the motor neuron-specific D42 driver were crossed to the indicated RNAi lines. Climbing potential of male (left) and female (right) progeny reared at 29°C was measured 7 days after eclosion as described in Materials and methods. Data analysis was performed using ordinary one-way ANOVA. Data are shown as mean ± SEM. n>100 flies, ****p≤0.0001. (**B**) Neuromuscular junction (NMJ) morphology analysis of

*Figure 4 continued on next page*

*Figure 4 continued*

D42>UBQLN2[4XALS] larvae expressing the indicated shRNAs. NMJs dissected from third instar larvae were stained with α-HRP and α-CSP antibodies and imaged by confocal microscopy. Scale bar: 10µm. (**C**) Number of NMJs harboring indicated phenotypes were tabulated from greater than 50 NMJs per genotype. Unpaired t-test was used for statistical analysis. Data are shown as mean ± SEM. (**D**) Pan-neuronal *Unc-5* knockdown enhances lifespan of Elav>UBQLN2[4XALS] flies. Recombinant Elav>UBQLN2[4XALS] or a Elav>Gal4 flies were crossed to the indicated RNAi lines (shLuci or shUnc5). Lifespan of male (left panel) and female (right panel) progeny reared at 27°C was measured as described in Materials and methods. n>50 flies.

The online version of this article includes the following source data for figure 4:

**Source data 1.** Data sets corresponding to *Figure 4A, C, D*.

(*Figure 6—figure supplement 1D*). These findings support the existence of endolysosomal defects in UBQLN2[ALS] iPSCs.

## UBQLN2[ALS] iMNs exhibit axonal inclusions and neurite defects

A failure to detect UBQLN2[ALS] aggregates in untreated iPSCs could be due to continuous cytosolic dilution of UBQLN2 during mitotic cell division or may reflect the absence of neuron-specific stimuli that promote UBQLN2 aggregation. To explore these ideas, we carried out immunostaining experiments using UBQLN2[WT], UBQLN2[P497H], or UBQLN2[4XALS] iMNs (*Figure 7A*). UBQLN2[WT], UBQLN2[P497H] showed largely diffuse localization patterns, whereas UBQLN2[4XALS] formed discrete aggregates distributed throughout iMN cell bodies and axons (*Figure 7A*). UBQLN2 aggregation was also observed in iMNs differentiated from an independently generated UBQLN2[4XALS] iPSC clone (Clone 2, *Figure 7—figure supplement 1A*).

F-actin labeling with phalloidin revealed that UBQLN2[WT] and UBQLN2[P497H] were evenly distributed throughout growth cone lamellipodia and filopodia. Lamellipodial UBQLN2[4XALS] staining was generally weaker, except for occasional brightly staining aggregates that were observed in ~50% of growth cones examined (*Figure 7B*). Notably, the UBQLN2[4XALS] mutation did not significantly impact the localization of DCC, which was highly enriched in filopodial spikes in iMNs of all UBQLN2 genotypes (*Figure 7—figure supplement 2*).

We evaluated neurite length and complexity in UBQLN2[WT], UBQLN2[P497H], and UBQLN2[4XALS] iMNs using Sholl analyses according to the diagram shown in *Equation 1* and *Figure 7C*, respectively. UBQLN2[4XALS] iMNs exhibited a significant reduction in average total neurite length, primary neurite length, and neurite branch length relative to UBQLN2[WT] iMNs, which manifested as an increased rate of Sholl decay (*Figure 7D*). By contrast, neurite length and complexity were comparable between UBQLN2[P497H] and UBQLN2[WT] iMNs, indicating the clinical P497H mutation is insufficient to disrupt neurite growth dynamics (*Figure 7D*).

Given that autophagy inhibition with BafA1 induced UBQLN2[4XALS] aggregation in iPSCs, we carried out similar studies in UBQLN2[WT], UBQLN2[P497H], and UBQLN2[4XALS] iMNs. BafA1 strongly increased the size and number of UBQLN2[4XALS] aggregates while having minimal effects on UBQLN2[WT] or UBQLN2[P497H] localization. BafA1-induced UBQLN2[4XALS] aggregates were observed in two different iMN clones and were strongly colocalized with the autophagy receptor, p62 (*Figure 8A, B and D*). In addition, UBQLN2[4XALS] aggregates often colocalized with, or were adjacent to, LAMP1-positive lysosomes (*Figure 8C and D*). By contrast, UBQLN2[4XALS] only weakly colocalized with endosomal Rab5 or the autophagosome marker, LC3 (*Figure 8C and D*). These findings suggest that the autophagy pathway suppresses endogenous UBQLN2[4XALS] aggregation in iMNs.

## UNC5B and DCC silencing reduce neurite and growth cone defects in UBQLN2[4XALS] iMNs

Mammals harbor a single *fra* ortholog, *DCC*, and four closely related paralogs with homology to *Unc-5*: *UNC5A*, *UNC5B*, *UNC5C*, and *UNC5D*. Among these, we focused on *UNC5B*, which is widely expressed in nervous tissue and has well-described roles in axon guidance and apoptosis regulation (*Wang et al., 2009*; *Ahn et al., 2020*; *Pradella et al., 2021*; *Tang et al., 2008*). To assess contributions of DCC and UNC5B signaling to UBQLN2-associated toxicity, we transduced UBQLN2[4XALS] iPSCs with lentiviral shRNA vectors targeting DCC or UNC5B and differentiated the cells into iMNs for neurite analysis. qPCR confirmed that expression of UNC5B and DCC was reduced ~40–60% in their respective shRNA-transduced iPSCs relative to iPSCs transduced with an NT shRNA vector (*Figure 9—figure*

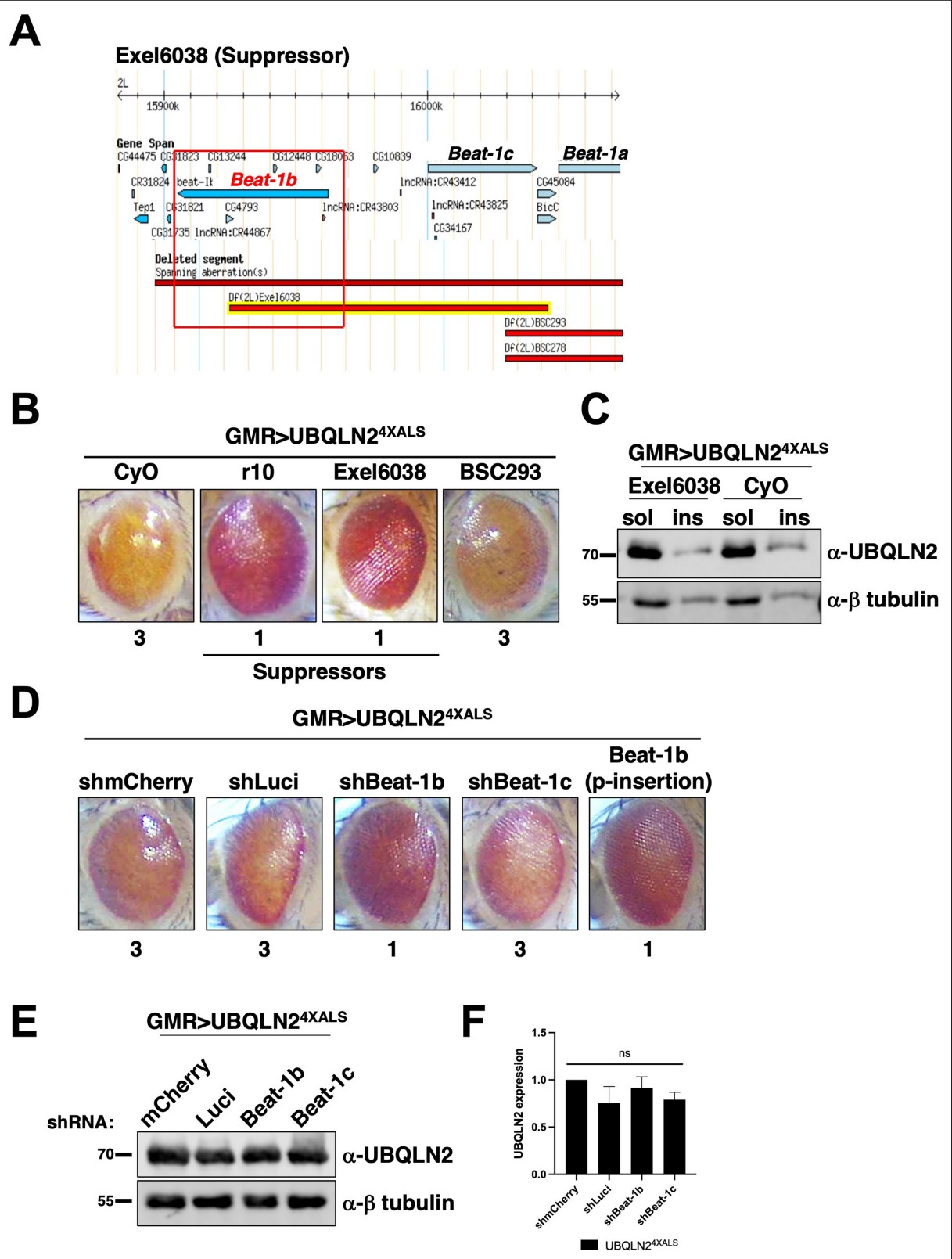

**Figure 5.** The axon guidance gene *beat-1b* is a UBQLN2[4XALS] suppressor. (**A**) Schematic of the *beat-1b* gene locus. (**B**) Representative eye phenotypes of GMR>UBQLN2[4XALS] flies harboring the indicated deficiency (Df) alleles. (**C**) UBQLN2 expression levels and RIPA solubility in whole heads of GMR>UBQLN2[4XALS]/Exel6038 and GMR>UBQLN2[4XALS]/CyO flies. (**D**) Representative eye phenotypes of GMR>UBQLN2[4XALS] flies expressing the indicated shRNAs. Eye degeneration scores are displayed below each eye image (**B, D**). (**E, F**) Knockdown of Beat-1b or Beat-1c does not inhibit UBQLN2

*Figure 5 continued on next page*

eLife Research article

Cell Biology | Neuroscience

*Figure 5 continued*

expression in GMR>UBQLN2$^{4XALS}$ flies. (**F**) Quantification of UBQLN2 expression normalized to β-tubulin. The bars represent mean with *SEM* of triplicate samples. Unpaired t-test was used for statistical analysis. .

The online version of this article includes the following source data for figure 5:

**Source data 1.** Uncropped Western blot images corresponding to *Figure 5C, E, F*.

*supplement 1A, B*). UBQLN2$^{4XALS}$ iMNs expressing *DCC* shRNA also showed reduced DCC immunoreactivity at filopodial spikes (*Figure 9—figure supplement 1C*). Both *UNC5B* and *DCC* knockdown significantly increased average total neurite length, primary neurite length, and branch length in UBQLN2$^{4XALS}$ iMNs relative to UBQLN2$^{4XALS}$ iMNs expressing a non-targeting (NT) control shRNA (*Figure 9A*). Neither *UNC5B* nor *DCC* silencing affected the size or number of UBQLN2$^{4XALS}$ aggregates (*Figure 9—figure supplement 2*), suggesting they influence toxicity pathway(s) downstream of UBQLN2 aggregation.

We next evaluated growth cone morphology in wild-type and UBQLN2$^{ALS}$ iMNs stained with phalloidin. Compared to UBQLN2$^{WT}$ or UBQLN2$^{P497H}$ iMNs, UBQLN2$^{4XALS}$ iMNs showed a high proportion of blunt-end termini versus growth cone termini, suggesting a defect in growth cone elaboration (*Figure 9B and D*). A preponderance of blunt-end termini was also observed in UBQLN2$^{4XALS}$ (Clone 2) iMNs, suggesting they are a specific consequence of the 4XALS mutation (*Figure 7—figure supplement 1B*). By contrast, UBQLN2$^{4XALS}$ iMNs expressing UNC5B or DCC shRNAs exhibited supernumerary growth cones along primary and secondary neurites and an increase in growth cone size and abundance relative to blunt-end termini (*Figure 9C and D*). These findings suggest that aberrant DCC-UNC5 signaling suppresses growth cone elaboration in UBQLN2$^{4XALS}$ iMNs and that axon guidance defects contribute to toxicity phenotypes in fly and iMN models for UBQLN2-associated ALS.

## Discussion

In this study we investigated how ALS-associated mutations in the Ub chaperone UBQLN2 cause cellular toxicity in *Drosophila* and gene-edited iMNs. Our findings support roles for endolysosomal dysfunction and axonal guidance defects as disease drivers in UBQLN2-associated ALS (*Figure 10*).

A genetic Df screen in *Drosophila* identified 35 loci (7 suppressors and 28 enhancers) that influenced toxicity of UBQLN2$^{ALS}$ mutants. Key to the identification of phenotypic suppressors was a UBQLN2$^{4XALS}$ allele that showed elevated HS-dependent toxicity relative to UBQLN2$^{ALS}$ point mutants, whose toxicities were difficult to discern from overexpressed UBQLN2$^{WT}$ (*Figures 1B and 2D*). The reason for enhanced toxicity of UBQLN2$^{4XALS}$ is unclear; however, its enhanced aggregation potential may overwhelm cellular proteostasis machinery and/or accelerate disease mechanisms that are slow to manifest in neurons harboring ALS point mutations. This is consistent with the fact that UBQLN2$^{4XALS}$ toxicity in flies was unmasked by HS, which is a well-known inducer of proteotoxicity. While caution must be taken when interpreting experiments employing the UBQLN2$^{4XALS}$ allele, it may serve as useful discovery tool for pathway identification in UBQLN2-associated ALS.

Among the 28 enhancer loci, we successfully mapped *Rab5* as the causal gene in BSC37. Rab5 silencing caused a hyperpigmented eye phenotype in flies expressing either wild-type or ALS-mutant UBQLN2 alleles, suggesting that overexpressed UBQLN2 proteins interfere with endosomal function. This finding is congruent with recent work by Senturk et al. describing a role for endogenous *dUbqln* in endolysosomal acidification (*Şentürk et al., 2019*). While we attempted to map culprit genes in several other UBQLN2 enhancer loci, we were unable to identify candidate genes whose silencing fully replicated the degenerative eye phenotypes seen with the Df crosses. A plausible explanation for this is that the disruption of multiple genes is responsible for the enhancer effects of some Df lines.

Our screen identified three UBQLN2$^{4XALS}$ suppressors (*lilli*, *Unc-5*, and *beat-1b*), while a causal gene responsible for the strong phenotypic rescue by BSC19 (*Figure 2C*) could not be mapped. *lilli* (*Figure 2—figure supplement 2*) is orthologous to mammalian *AFF2/FMR2* (*fragile X mental retardation 2*), which plays a role in transcriptional elongation (*Luo et al., 2012*; *Wittwer et al., 2001*). Interestingly, *lilli* was also identified as a phenotypic suppressor in *Drosophila* models for TDP-43 and C9ORF72-associated ALS (*Yuva-Aydemir et al., 2019*; *Berson et al., 2019*). In the C9ORF72 model, *lilli* mutations suppressed expression and toxicity of overexpressed poly(GR) DPRs that are produced via repeat-associated, non-ATG-dependent translation of a G4C2 hexanucleotide repeat expansion

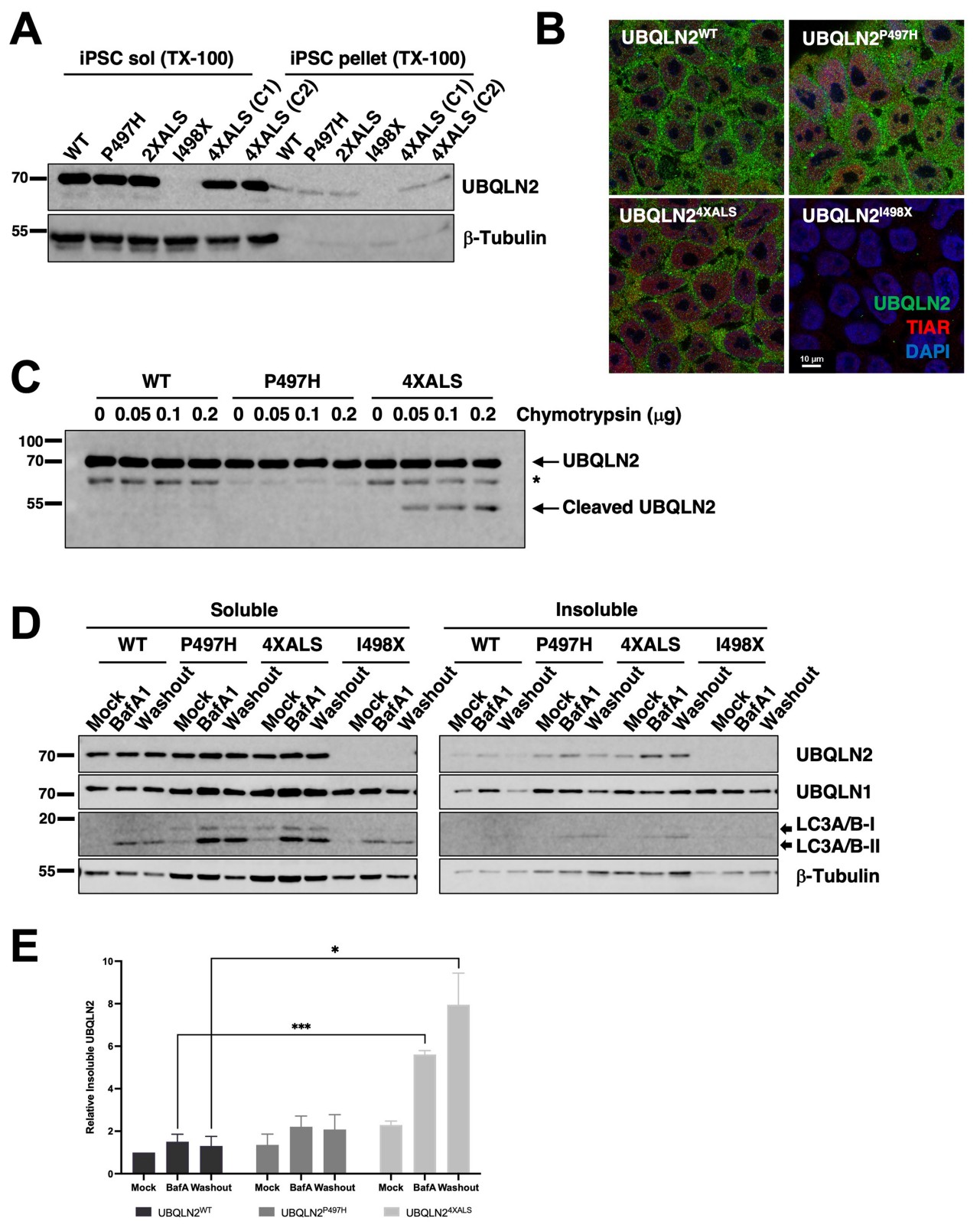

**Figure 6.** Localization and solubility of UBQLN2[ALS] mutants in induced pluripotent stem cells (iPSCs). (**A**) Extracts from UBQLN2[WT], UBQLN2[P497H], UBQLN2[2XALS], UBQLN2[I498X], or UBQLN2[4XALS] iPSCs (clone 1 [C1] and clone 2 [C2]) were separated into soluble and insoluble fractions in 1% Triton X-100 (TX-100) buffer and immunoblotted with α-UBQLN2, and α-β-tubulin antibodies. (**B**) Localization patterns of wild-type and UBQLN2[ALS] proteins in iPSCs. UBQLN2[WT], UBQLN2[P497H], UBQLN2[4XALS], and UBQLN2[I498X] iPSCs were stained with α-UBQLN2 and α-TIAR antibodies and imaged by confocal

*Figure 6 continued on next page*

Figure 6 continued

microscopy. Note the lack of cytosolic aggregates. (**C**) Cell extracts from iPSCs of the indicated genotypes were incubated at room temperature with increasing amounts of chymotrypsin for 5 min. After separation by SDS-PAGE, the proteins were immunoblotted with α-UBQLN2 antibodies. Positions of full-length and cleaved UBQLN2 are denoted by arrows. *: non-specific band. (**D**) Autophagy inhibition with BafA1 reduced solubility of endogenous UBQLN2ALS proteins. UBQLN2WT, UBQLN2P497H, UBQLN24XALS, and UBQLN2I498X iPSCs were treated with 100 nM of BafA1 for 16 hr followed by BafA1 washout and incubation in BafA1-free growth media for 8 hr. Detergent extracts were separated into soluble and insoluble fractions and analyzed by SDS-PAGE and immunoblotting using UBQLN2, UBQLN1, LC3A/B, and β-tubulin antibodies. (**E**) Quantification of UBQLN2 solubility in UBQLN2WT, UBQLN2P497H, and UBQLN24XALS iPSCs from (**D**). The bars represent mean with *SEM* of triplicate samples. Unpaired t-test was used for statistical analysis. *p≤0.05, ***p≤0.001.

The online version of this article includes the following source data and figure supplement(s) for figure 6:

**Source data 1.** Uncropped Western blot images corresponding to *Figure 6A, C, D, E*.

**Figure supplement 1.** Endogenous UBQLN2ALS mutants perturb lysosomes.

**Figure supplement 1—source data 1.** Data sets corresponding to *Figure 6—figure supplement 1C, D*.

(HRE) in the *C9ORF72* gene (*Van't Spijker and Almeida, 2023*). *lilli/FMR2* silencing also reduced transcription of G4C2-containing repeats in iMNs and partially reversed neurite defects in patient-derived iMNs harboring the *C9ORF72* HRE (*Yuva-Aydemir et al., 2019*; *Berson et al., 2019*). UBQLN24XALS protein levels were not affected by a *lilli* LOF allele in *Drosophila* (*Figure 2—figure supplement 2*), suggesting that transcriptional suppression is not responsible for phenotypic rescue in our studies. Alternatively, because *lilli* was also identified as a suppressor of *hairless* overexpression toxicity in the fly eye (*Müller et al., 2005*), *lilli* LOF alleles may suppress overlapping death pathways engaged by neurodegeneration-associated proteins.

Our findings suggest that axon guidance pathways play an important role in UBQLN2-mediated toxicity in *Drosophila*. Unc-5 fulfills dual functions as a repulsive axon guidance factor and neuronal dependence receptor (*Boyer and Gupton, 2018*; *Labrador et al., 2005*; *Keleman and Dickson, 2001*), and either or both functions could underlie phenotypic rescue of UBQLN24XALS flies by *Unc-5* silencing. Knockdown of *fra*, which mediates chemoattractive responses to netrin, also partially rescued UBQLN24XALS-associated eye and motor neuron phenotypes (*Figures 3F, 9A, C and D*), raising the possibility that heterodimeric Unc-5-Fra complexes mediate toxicity initiated by UBQLN2ALS mutants. Alternatively, Unc-5 and Fra/DCC may function in partially redundant fashion to instigate toxicity in UBQLN2ALS flies.

The identification of *beat-1b* as a UBQLN2 modifier further supports axon pathfinding defects as a disease driver in the UBQLN24XALS flies. Beat proteins mediate guidance of motor neuron axons through transient interactions with transmembrane Sidestep receptors whose expression pattern on muscle, muscle stem cells, and neurons constitutes a stereotypic guidance path (*Fambrough and Goodman, 1996*; *Siebert et al., 2009*; *Aberle, 2009*; *Arzan Zarin and Labrador, 2019*; *Li et al., 2017*). Although no clear human ortholog exists, Beats exhibit weak homology to mammalian NekL3/SynCAM2/CADM2, a nectin-like molecule that mediates adhesion of myelinated axons and oligodendrocytes (*Frei et al., 2014*; *Biederer et al., 2002*). The potential roles of nectin-like molecules in the toxicity of UBQLN2ALS mutants in human neurons remain to be determined.

We developed UBQLN2 gene-edited iPSCs and iMNs to investigate UBQLN2 pathomechanisms at the cellular level. Endogenous UBQLN2P497H and UBQLN24XALS exhibited wild-type localization and solubility in iPSCs in the absence of stress; however, the relative insolubility of UBQLN24XALS seen in overexpression studies was unmasked when iPSCs were treated with the lysosomal acidification inhibitor BafA1 (*Figure 6—figure supplement 1*). UBQLN24XALS iMNs exhibited constitutive, p62-positive, aggregates that were distributed throughout the soma, axon, and growth cone lamellipodia, raising the possibility that such aggregates interfere with growth cone dynamics (*Figures 7 and 8*). Indeed, UBQLN24XALS iMNs exhibited reduced neurite length, diminished neurite complexity, and reduced growth cone numbers relative to wild-type and UBQLN2P497H iMNs (*Figures 7 and 9*). While these findings imply that UBQLN24XALS toxicity is tightly linked to its aggregation potential, it remains possible that soluble forms of UBQLN24XALS and clinical UBQLN2ALS proteins also disrupt key cellular processes. Given genetic interactions between UBQLN2 and Rab5 in flies (*Figure 2—figure supplement 1*) and lysosomal enlargement seen in UBQLN2P497H and UBQLN24XALS iPSCs (*Figure 6—figure supplement 1*), the endolysosomal pathway may be a particularly relevant site of soluble UBQLN2 toxicity.

Knockdown of *UNC5B* and *DCC* partially reversed the neurite complexity and growth cone morphology defects of UBQLN2[4XALS] iMNs, indicating that the UBQLN2[4XALS] toxicity mechanism is at least partially conserved between flies and humans (***Figure 9***). As in *Drosophila*, mammalian DCC and UNC5 family receptors have been implicated in both axon guidance and apoptosis regulation (***Boyer and Gupton, 2018***; ***Llambi et al., 2005***; ***Wang et al., 2009***; ***Ahn et al., 2020***; ***Pradella et al., 2021***; ***Tang et al., 2008***; ***Ackerman et al., 1997***; ***Leonardo et al., 1997***; ***Barnault et al., 2018***; ***Tanikawa et al., 2003***; ***Williams et al., 2006***; ***Guenebeaud et al., 2010***; ***Miyamoto et al., 2010***). UNC5 and DCC/Fra harbor extended cytoplasmic domains that regulate caspase activation and are themselves targets for caspase-mediated cleavage (***Wang et al., 2009***, ***Llambi et al., 2001***; ***Forcet et al., 2001***). Engagement by netrin ligand is thought to suppress the intrinsic apoptotic potential of the UNC5 death domain (***Pradella et al., 2021***; ***Wang et al., 2009***), while γ-secretase-mediated cleavage of UNC5C has been linked to neuronal apoptosis in Alzheimer's disease (AD) (***Chen et al., 2021***). The contributions of UNC5 death signaling to toxicity phenotypes in UBQLN2[ALS] iMNs and flies await future study.

Distal axon and synaptic defects are implicated in ALS pathogenesis and have been observed in diverse ALS disease models. For instance, reduced expression of the microtubule binding protein Stathmin 2 due to misregulation of its RNA splicing and/or polyadenylation is strongly linked to axonal growth and regeneration defects in TDP-43-associated ALS (***Melamed et al., 2019***; ***Klim et al., 2019***). Single nucleotide polymorphisms that promote inclusion of a cryptic cassette exon in the presynaptic regulator *UNC13A* are a risk factor for fALS (***van Es et al., 2009***), whereas loss of nuclear TDP-43 has been linked to UNC13A missplicing and synaptic defects in sALS (***Ma et al., 2022***; ***Brown et al., 2022***). While *UNC5B* has not been implicated as an ALS gene, signaling downstream of UNC5 may contribute to axonal retraction and/or synaptic phenotypes in neurodegenerative disease. Consistent with this notion, *UNC5B* has been linked to neurodegeneration in the 6-OHDA model of Parkinson's disease (***Jasmin et al., 2021***) and *UNC5C* has been nominated as a risk allele in late-onset AD (***Wetzel-Smith et al., 2014***; ***Korvatska et al., 2015***; ***Li et al., 2018***). Future studies will define the contributions of pathologic UNC5 signaling to the development and/or progression of ALS/FTD.

Finally, while findings in flies and iMNs support a conserved role for axon guidance defects in the UBQLN2 toxicity mechanism, there are several limitations to our study. First, despite careful attempts to focus on mutation-specific phenotypes, overexpression may elicit disease non-specific toxicities in flies, with subsequent impacts on genetic screens. Second, the UBQLN2[4XALS] mutant is not a bona fide disease allele and may elicit toxicities unrelated to those caused by clinical ALS mutations. Third, it is possible that axon guidance genes are most relevant to UBQLN2 toxicity in the context of the developing nervous system. Finally, neonatal iPSCs and their derivative iMNs, while possessing numerous strengths, are unlikely to capture age-dependent abnormalities that contribute to neurodegeneration in an intact human nervous system.

## Materials and methods

**Key resources table**

| Reagent type (species) or resource | Designation | Source or reference | Identifiers | Additional information |
|---|---|---|---|---|
| Genetic reagent (*Drosophila melanogaster*) | Deficiency | BDSC | BDSC7521 | w[1118]; Df(2L)Exel6038, P{w[+mC]=XPU}Exel6038/CyO |
| Genetic reagent (*D. melanogaster*) | *beat-1c*/RNAi | BDSC | BDSC64528 | y(1) sc[*] v(1) sev(21); P{y[+t7.7] v[+t1.8]=TRiP.HMC05547}attP40 |
| Genetic reagent (*D. melanogaster*) | *beat-1b*/RNAi | BDSC | BDSC55938 | y(1) sc[*] v(1) sev(21); P{y[+t7.7] v[+t1.8]=TRiP.HMC04226}attP40 |
| Genetic reagent (*D. melanogaster*) | *Rab5*/RNAi | BDSC | BDSC34832 | y(1) sc[*] v(1) sev(21); P{y[+t7.7] v[+t1.8]=TRiP.HMS00147}attP2 |
| Genetic reagent (*D. melanogaster*) | Luciferase/RNAi | BDSC | BDSC31603 | y(1) v(1); P{y[+t7.7] v[+t1.8]=TRiP.JF01355}attP2 |

*Continued on next page*

*Continued*

| Reagent type (species) or resource | Designation | Source or reference | Identifiers | Additional information |
|---|---|---|---|---|
| Genetic reagent (*D. melanogaster*) | *Rab5*/RNAi | BDSC | BDSC30518 | y(1) v(1); P{y[+t7.7] v[+t1.8]=TRiP.JF03335}attP2 |
| Genetic reagent (*D. melanogaster*) | *beat-1b*/P element | BDSC | BDSC18802 | w[1118]; PBac{w[+mC]=WH}beat-Ib[f04746] |
| Genetic reagent (*D. melanogaster*) | Rab5/overexpression | BDSC | BDSC43336 | w[*]; P{w[+mC]=UAS-GFP-Rab5}3 |
| Genetic reagent (*D. melanogaster*) | *Unc-5*/RNAi | BDSC | BDSC33756 | y(1) sc[*] v(1) sev(21); P{y[+t7.7] v[+t1.8]=TRiP.HMS01099}attP2 |
| Genetic reagent (*D. melanogaster*) | *Unc-5*/RNAi | VDRC | VDRC8138 | GD RNAi |
| Genetic reagent (*D. melanogaster*) | *Unc-5*/RNAi | VDRC | VDRC110155 | KK RNAi |
| Genetic reagent (*D. melanogaster*) | GFP/overexpression | BDSC | BDSC5430 | w[1118]; P{w[+mC]=UAS-EGFP}34/TM3, Sb(1) |
| Genetic reagent (*D. melanogaster*) | Deficiency | BDSC | BDSC24370 | w[1118]; Df(2R)BSC346/CyO |
| Genetic reagent (*D. melanogaster*) | mCherry/RNAi | BDSC | BDSC35787 | y(1) sc[*] v(1) sev(21); P{y[+t7.7] v[+t1.8]=UAS-mCherry.VALIUM10}attP2 |
| Genetic reagent (*D. melanogaster*) | *frazzled*/RNAi | BDSC | BDSC31469 | y(1) v(1); P{y[+t7.7] v[+t1.8]=TRiP.JF01231}attP2 |
| Genetic reagent (*D. melanogaster*) | *frazzled*/RNAi | BDSC | BDSC31664 | y(1) v(1); P{y[+t7.7] v[+t1.8]=TRiP.JF01457}attP2 |
| Genetic reagent (*D. melanogaster*) | *frazzled*/RNAi | BDSC | BDSC40826 | y(1) sc[*] v(1) sev(21); P{y[+t7.7] v[+t1.8]=TRiP.HMS01147}attP2 |
| Genetic reagent (*D. melanogaster*) | Deficiency (*lilli*) | BDSC | BDSC9610 | BSC180 |
| Genetic reagent (*D. melanogaster*) | Deficiency (*Rab5*) | BDSC | BDSC7144 | BSC37 |
| Genetic reagent (*D. melanogaster*) | Deficiency (*lilli*) | BDSC | BDSC94697 | ED4651 |
| Genetic reagent (*D. melanogaster*) | Deficiency (*lilli*) | BDSC | BDSC99 | C144 |
| Genetic reagent (*D. melanogaster*) | *Lilli* (LOF allele) | BDSC | BDSC5726 | *lilli*[A17-2] cn(1) bw(1)/CyO |
| Genetic reagent (*D. melanogaster*) | *Unc-5* (LOF) | Greg Bashaw; ***Labrador et al., 2005*** | https://doi.org/10.1016/j.cub.2005.06.058 | *Unc-5*[3] |
| Genetic reagent (*D. melanogaster*) | *Unc-5* (LOF) | Greg Bashaw; ***Labrador et al., 2005*** | https://doi.org/10.1016/j.cub.2005.06.058 | *Unc-5*[8] |
| Genetic reagent (*D. melanogaster*) | Unc-5/overexpression | Greg Bashaw | | HA-Unc-5 (Chr2) |
| Genetic reagent (*D. melanogaster*) | Unc-5/overexpression | Greg Bashaw | | HA-Unc-5 (Chr3) |
| Cell line (*Homo sapiens*) | UBQLN2[WT] | WC031i-5907–6 | WT | |
| Cell line (*Homo sapiens*) | UBQLN2[P497H] | This paper | P497H | Mutation using CRISPR |

*Continued on next page*

*Continued*

| Reagent type (species) or resource | Designation | Source or reference | Identifiers | Additional information |
|---|---|---|---|---|
| Cell line (*Homo sapiens*) | UBQLN2$^{2XALS}$ | This paper | P497H, P525S | Mutation using CRISPR |
| Cell line (*Homo sapiens*) | UBQLN2$^{4XALS}$ | This paper | P497H, P506T, P509S, P525S | Mutation using CRISPR |
| Cell line (*Homo sapiens*) | UBQLN2$^{I498X}$ | This paper | I498X | Mutation using CRISPR |
| Transfected construct (humani PSCs) | shUNC5B | Sigma-Aldrich | TRCN0000442978 | Lentiviral construct to transfect and express the shRNA in iPSCs |
| Transfected construct (human iPSCs) | shDCC | Sigma-Aldrich | TRCN0000010318 | Lentiviral construct to transfect and express the shRNA in iPSCs |
| Antibody | Anti-UBQLN2 (Mouse monoclonal antibody) | Abcam | Cat#: Ab190283 | IF (1:1000), WB (1:10000) |
| Antibody | Anti-UBQLN2 (Rabbit polyclonal antibody) | Cell Signaling Technology | Cat#: 85509 | IF (1:500), WB (1:2000) |
| Antibody | Anti-β-Tubulin (Mouse monoclonal antibody) | EMD Millipore | Cat#: 05–661 | WB (1:2000) |
| Antibody | Anti-LC3A/B (Rabbit polyclonal antibody) | Cell Signaling Technology | Cat#: 12741S | IF (1:500), WB (1:1000) |
| Antibody | Anti-LAMP1 (Mouse monoclonal antibody) | Santa Cruz Biotechnology | Cat#: sc-20011 | IF (1:500), WB (1:1000) |
| Antibody | Anti-Rab5A (Rabbit polyclonal antibody) | Cell Signaling Technology | Cat#: 46449S | IF (1:500), WB (1:1000) |
| Antibody | Anti-UBQLN1 (Rabbit polyclonal antibody) | Cell Signaling Technology | Cat#: 14526 | IF (1:500), WB (1:2000) |
| Antibody | Anti-Tuj1 (Mouse monoclonal antibody) | EMD Millipore | Cat#: MAB1637MI | IF (1:500) |
| Antibody | Anti-DCC (Rabbit polyclonal antibody) | Invitrogen | Cat#: PA5-50946 | IF (1:500) |
| Antibody | Anti-DLG (Mouse monoclonal antibody) | DSHB | Cat#: 4F3 | IF (1:100) |
| Antibody | Anti-HRP- Cy3-conjugated (Goat polyclonal antibody) | Jackson ImmunoResearch | Cat#:123-165-021 | IF (1:100) |
| Sequence-based reagent | GAPDH-F | Sigma-Aldrich | qPCR primers | GTCTCCTCTGACTTCAACAGCG |
| Sequence-based reagent | GAPDH-R | Sigma-Aldrich | qPCR primers | ACCACCCTGTTGCTGTAGCCAA |
| Sequence-based reagent | UNC5B-F | Sigma-Aldrich | pPCR primers | ACTGCCGTGACTTCGACAC |
| Sequence-based reagent | UNC5B-R | Sigma-Aldrich | qPCR primers | GCCTTGCCGTCTTAAAGTTGA |
| Sequence-based reagent | DCC-F | Sigma-Aldrich | qPCR primers | GACTTTACCAATGTGAGGCATCT |
| Sequence-based reagent | DCC-R | Sigma-Aldrich | qPCR primers | GGTCCTGCTACTGCAACTTTT |
| Chemical compound, drug | CHIR99021 | Tocris | 4423 | Chemical compound, drug |

*Continued on next page*

*Continued*

| Reagent type (species) or resource | Designation | Source or reference | Identifiers | Additional information |
|---|---|---|---|---|
| Chemical compound, drug | DMH-1 | Tocris | 4126 | Chemical compound, drug |
| Chemical compound, drug | SB431542 | Stemgent | 04-0010 | Chemical compound, drug |
| Chemical compound, drug | Retinoic acid | Stemgent | 04-0021 | Chemical compound, drug |
| Chemical compound, drug | Purmorphamine | Stemgent | 04-0009 | Chemical compound, drug |
| Chemical compound, drug | Compound E | EMD Millipore | 565790 | Chemical compound, drug |
| Software, algorithm for RNA-Seq analysis | *Drosophila melanogaster* genome (dmel-all-chromosome-r6.27, FlyBase) | https://github.com/ENCODE-DCC/rna-seq-pipeline; *ENCODE DCC, 2022* STAR 2.7.1a | DESeq2, MetaScape website | |
| Software, algorithm for mass spectrometry analysis | MetaMorpheus software program | | FlashLFQ | |

## *Drosophila* methods

Flies were maintained with the standard cornmeal-yeast medium (Nutri-Fly BF #66-112, Genesee Scientific) supplemented with propionic acid and all crosses were performed at 22°C. For HS experiments, all crosses were performed at indicated temperatures (27°C or 29°C). Note: the UAS-Gal4 promoter contains heat-shock elements that increase transgene expression at 27°C and 29°C. Generation of isogenic UAS-UBQLN2 stocks using PhiC31 integration was previously described.

## RNA-Seq and gene expression

Total RNA was isolated from 100 male fly heads using the TRIzol reagent (Invitrogen, 15596018) following the manufacturer's protocol and treated with TURBO Dnase (Invitrogen, AM2239). RNA samples were prepared with three biological replicates for each genotype and each temperature. Then RNA samples were sent to Novogene (Novogene Co., Ltd, Sacramento, CA, USA) for non-stranded cDNA library building and sequencing at PE150 with NovoSeq 6000. Raw reads adapters were trimmed by fastp (*Chen et al., 2018*) and then were mapped to *D. melanogaster* genome (dmel-all-chromosome-r6.27, FlyBase) by STAR with the setting suggested by ENCODE project (https://github.com/ENCODE-DCC/rna-seq-pipeline, *Dobin et al., 2013* STAR 2.7.1a). The number of RNA-Seq reads mapped to each transcript was summarized with featureCounts (*Liao et al., 2014*) and differential expression was called using DESeq2 (*Love et al., 2014*). The GO analysis was performed on MetaScape website (*Zhou et al., 2019*).

## Fly brain immunohistochemistry

The protocol for fly brain immunohistochemistry was adapted from a previously published protocol (*Zhan et al., 2013*). Adult fly brains were dissected using a pair of fine forceps in PBS (or 0.3% TX-100 in PBS), fixed, blocked with normal goat serum (NGS), and stained with primary antibodies at 1:500 dilution for two overnights at 4°C. After subsequent secondary antibody staining, DAPI was added for the nuclear staining. Images were acquired using a Nikon A1 confocal microscope using a 60× oil lens.

## Mass spectrometry

Fly lysates were prepared by homogenizing 100 male fly heads in 200 µl of lysis buffer containing 20 mM Tris-HCl (pH 8.0), 138 mM NaCl, 10 mM KCl, 1 mM MgCl₂, 1 mM EDTA, 0.5% sodium deoxycholate w/v, and 0.1% SDS w/v (RIPA buffer w/o NP-40). Samples were centrifuged at 20,000 × *g*

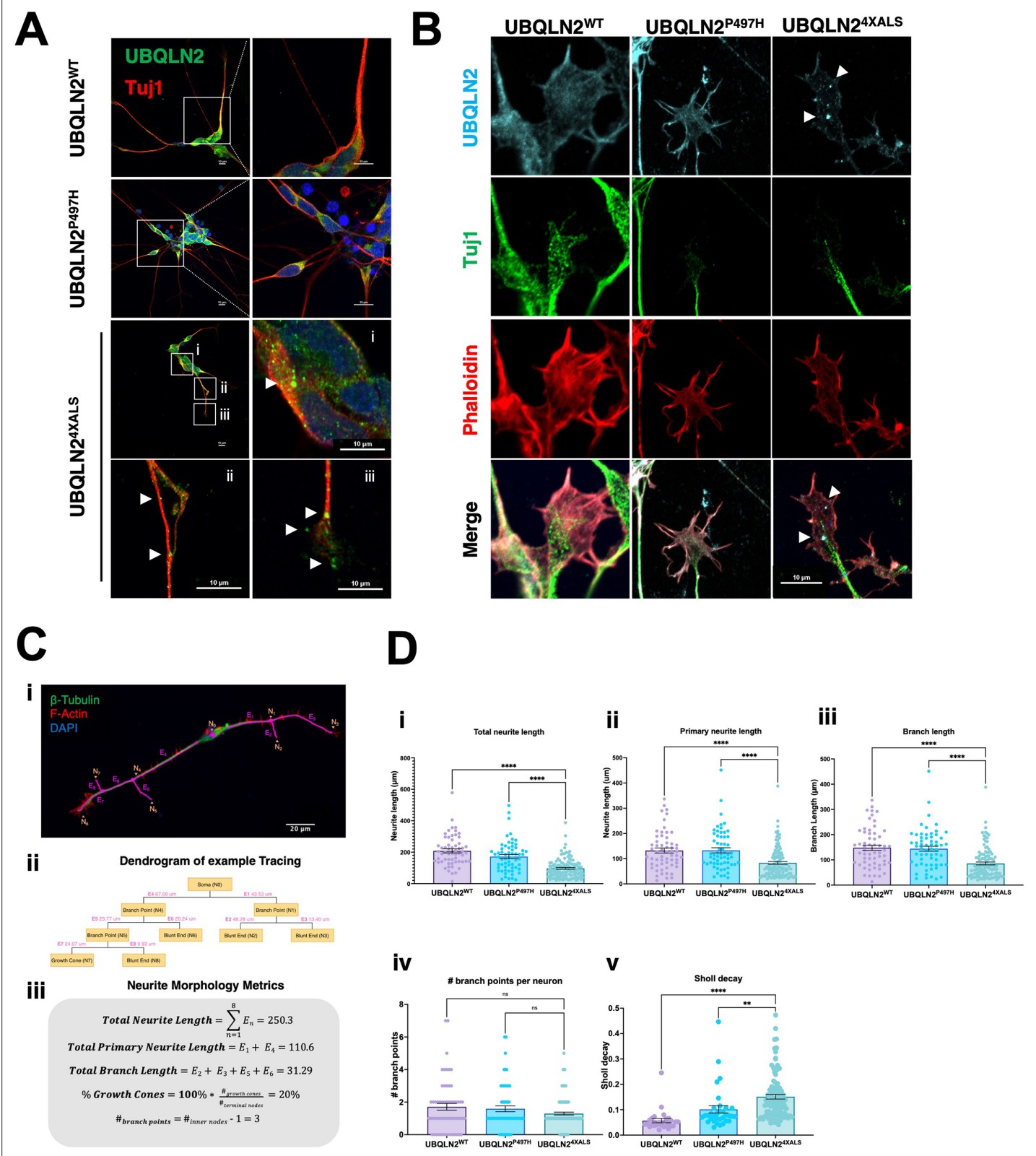

**Figure 7.** Protein aggregation and neurite defects in UBQLN2$^{ALS}$ inducible motor neurons (iMNs). (**A**) UBQLN2$^{WT}$, UBQLN2$^{P497H}$, and UBQLN2$^{4XALS}$ iMNs were immunostained for UBQLN2 and Tuj1. Magnified images of UBQLN2$^{4XALS}$ iMN soma (**i**), axons (ii), and neurite terminals (iii) are shown; aggregates are marked with arrows. (**B**) UBQLN2 localizes to growth cone lamellipodia and filopodia. Differentiated iMNs of the indicated genotypes were costained for UBQLN2, Tuj1, and filamentous actin (phalloidin). Note reduced complexity of the UBQLN2$^{4XALS}$ growth cone. Arrowheads indicate

*Figure 7 continued on next page*

*Figure 7 continued*

UBQLN2[4XALS] aggregates. Scale bars = 10 μm. (**C**) Schematic of Sholl analysis. Tracing example of an iMN (**i**) displays the paths representing individual neuron structure, with nodes ($N_i$) and edges ($E_j$) corresponding to the dendrogram on (**ii**) and (**iii**). (**D**) UBQLN2[4XALS] iMNs exhibit reduced complexity. UBQLN2[WT], UBQLN2[P497H], and UBQLN2[4XALS] iMNs were stained with α-Tuj1 and imaged by confocal microscopy. One hundred neurons of the indicated genotypes were traced using Simple Neurite Tracer (SNT) and subjected to Sholl image analysis to quantify total neurite projection path length (**i**), primary neurite length (**ii**), terminal neurite length (**iii**), neurite branch points (**iv**), and Sholl decay (**v**). Data analysis was performed using ordinary one-way ANOVA. Data are shown as mean ± SEM. n>100 iMNs, **p≤0.01, ****p≤0.0001.

The online version of this article includes the following source data and figure supplement(s) for figure 7:

**Source data 1.** Data sets corresponding to *Figure 7D*.

**Figure supplement 1.** Protein aggregation and reduced neurite complexity in an independent UBQLN2[4XALS] inducible motor neuron (iMN) line.

**Figure supplement 2.** The UBQLN2[4XALS] mutation did not significantly impact the localization of DCC.

for 10 min and soluble fractions subjected to tryptic digestion and orbitrap MS using the filter aided sample preparation method (*Wiśniewski et al., 2009*). We performed two technical replicates for each of the three biological replicates. The tryptic digest solution was desalted/concentrated using an Omix 100 μl (80 μg capacity) C18 tip and the peptides were analyzed by HPLC-ESI-MS/MS using a system consisting of a high-performance liquid chromatograph (nanoAcquity, Waters) connected to an electrospray ionization Orbitrap mass spectrometer (QE HF, Thermo Fisher Scientific). HPLC separation employed a 100×365 μm fused silica capillary micro-column packed with 20 cm of 1.7 μm diameter, 130 Å pore size, C18 beads (Waters BEH), with an emitter tip pulled to approximately 1 μm using a laser puller (Sutter Instrument). Peptides were loaded on-column at a flow rate of 400 nl/min for 30 min and then eluted over 120 min at a flow rate of 300 nl/min with a gradient of 5–35% acetonitrile, in 0.2% formic acid. Full-mass profile scans were performed in the FT orbitrap between 375 and 1500 m/z at a resolution of 120,000, followed by MS/MS HCD scans of the 10 highest intensity parent ions at 30% relative collision energy and 15,000 resolution, with a mass range starting at 100 m/z. Dynamic exclusion was enabled with a repeat count of one over a duration of 30 s. The MetaMorpheus software program was used to identify peptides and proteins in the samples (*Shortreed et al., 2015*; *Solntsev et al., 2018*). Protein fold changes were quantified by FlashLFQ (*Millikin et al., 2020*; *Yin et al., 2019*; *Millikin et al., 2018*).

## Genetic screening

We employed the Bloomington Deficiency Kit for chromosome 2 (DK2L and DK2R) comprised of 194 different lines. All 194 lines were crossed to GMR-Gal4/CyO or homozygous GMR>UBQLN2[P497H] or GMR>UBQLN2[4XALS] flies at 29°C. A minimum of 30 F1 progeny containing GMR>UBQLN2 either the Df chromosome or balancer chromosome were analyzed for eye morphology 1–3 days post eclosion using a blinded, 1–5 grading system, with a score of 1 representing a control (GMR-Gal4) eye; 3 corresponding to the unmodified UBQLN2[4XALS] phenotype at 29°C; and 5 representing severe eye degeneration featuring more than 50% necrotic tissue. We were unable to derive UBQLN2 progeny for a handful Df lines crossed to UBQLN2[4XALS], suggesting lethal genetic interactions. All putative modifier Dfs were retested in secondary screens that included side-by-side crosses to GMR-Gal4, GMR>UBQLN2[WT], GMR>UBQLN2[P497H], and GMR>UBQLN2[4XALS]. Those Dfs that were confirmed to modify GMR > UBQLN2 eye phenotypes in both screens were deemed bona fide modifier Dfs. Sexually dimorphic phenotypes were also scored. *Drosophila* eye pictures were acquired using Leica S9 i Stereomicroscope.

## *Drosophila* climbing assay

Climbing assay was modified from methods described previously (*Kim et al., 2018*). Climbing ability was measured by tapping ~10 flies to the bottom of a graduated testing vial (15 cm) and taking videos over of fly movement over the course of 10 s. More than 100 flies for each genotype and each gender were used for climbing ability. Video frames at the 5 s time point were used to record the position of each fly using the multi-point plugin in ImageJ. Using the final positions of every fly and respective starting points also marked with ImageJ's multi-point, the vertical displacement and velocity of every fly was calculated. Data showing velocity of each individual fly were graphed as scattered plots with

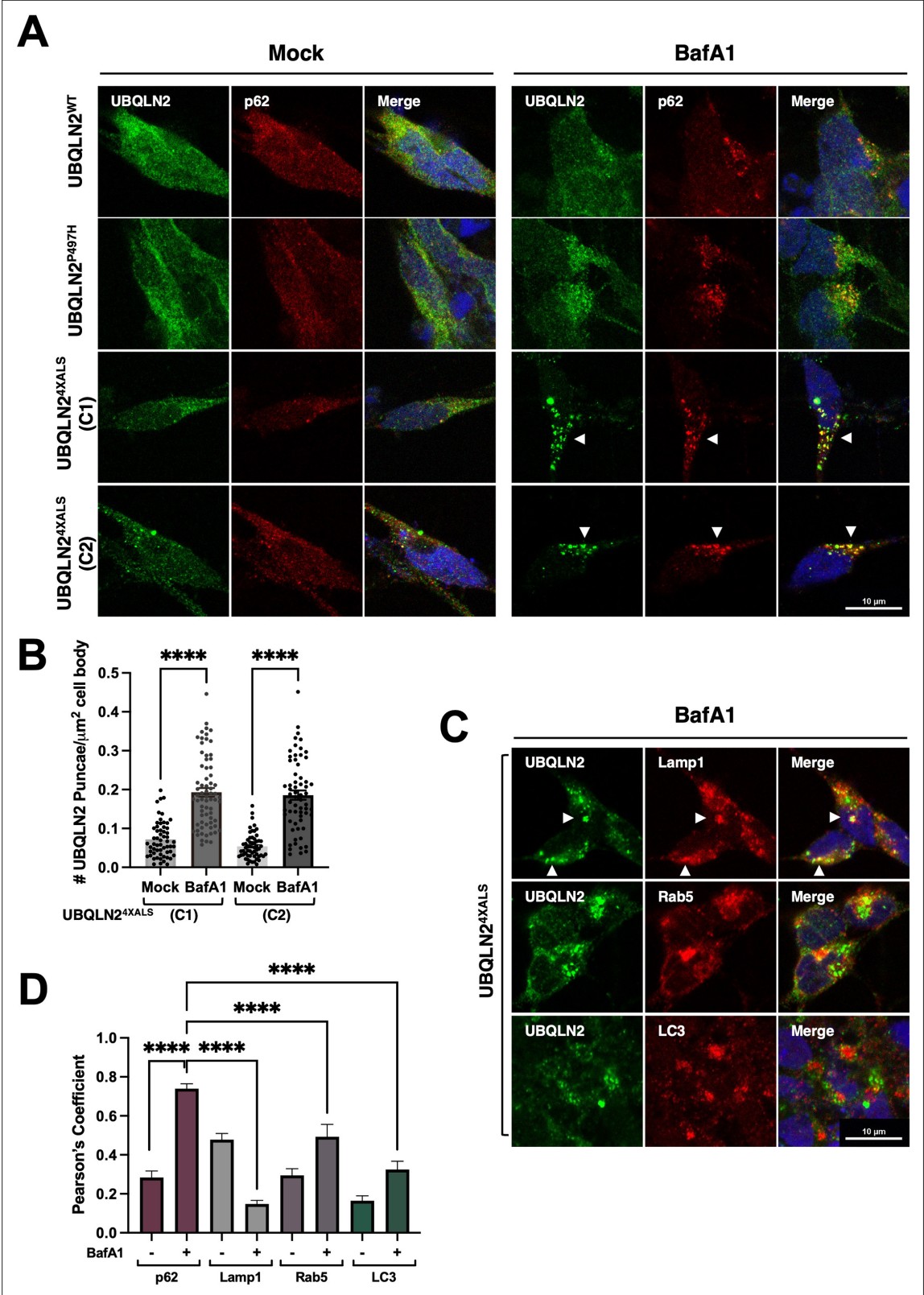

**Figure 8.** BafA1 induces p62-positive UBQLN2^4XALS aggresomes in inducible motor neurons (iMNs). (**A**) UBQLN2^WT, UBQLN2^P497H, and UBQLN2^4XALS iMNs were treated with BafA1 (50 nM) for 16 hr and processed for immunostaining with UBQLN2 and p62. Arrowheads indicate UBQLN2^4XALS aggregates that were colocalized with p62. (**B**) The number of UBQLN2^4XALS aggregates was analyzed on a per cell basis using Fiji. Data analysis was performed using ordinary one-way ANOVA. Data are shown as mean ± SEM. n>50 iMNs, ****p≤0.0001. (**C**) UBQLN2^4XALS iMNs were treated with BafA1 and processed for

*Figure 8 continued on next page*

*Figure 8 continued*

immunostaining with UBQLN2 and Lamp1, Rab5, or LC3. Arrowheads indicate UBQLN2[4XALS] colocalization with Lamp1. Scale bars = 10 μm. (**D**) Pearson's correlation coefficients for colocalization assays. Pearson's coefficients were plotted as a bar graph to compare the colocalization of UBQLN2 with p62, LAMP1, Rab5, or LC3. n=10 iMNs, Error bars represent SEM, ****p≤0.0001 (ordinary one-way ANOVA).

The online version of this article includes the following source data for figure 8:

**Source data 1.** Data sets corresponding to *Figure 8B, D*.

mean climbing distance and SEM. Unpaired t-test with Welch's correction were used for statistical analysis for different groups of flies.

## *Drosophila* NMJ assay

NMJ assay was modified from methods described previously (*Kim et al., 2018*). Third-instar, wandering larvae from the F1 generation were rinsed in ice-cold PBS (Lonza, 17512F) and dissected along the dorsal midline. All tissues except the brain and nerves were removed to expose the muscles and NMJs. The dissected larval pelt was fixed in 4% paraformaldehyde for 20 min at room temperature. The larval pelts were given a wash with PBS followed by blocking with 5% NGS in 0.1% PBST (0.1% TX-100 in PBS). Following blocking, the larval pelts were probed with primary antibodies overnight at 4°C. They were then washed several times with 0.1% PBST followed by incubation with secondary antibodies for 2 hr at room temperature, subsequently followed by washes with 0.1% PBST. Larvae were then mounted onto slides using Prolong Gold mounting media. Confocal images were acquired using Zeiss LSM 710 confocal microscope and a 60× oil objective was used to image the NMJs. Both primary and secondary antibody solutions were prepared in 5% NGS in 0.1% PBST. For primary antibodies, the following dilutions were used: 1:100 Cy3-conjugated goat anti-HRP (Jackson ImmunoResearch, 123-165-021); 1:100 mouse anti-DLG 4F3 (DSHB). For secondary antibodies, the following antibody dilutions were used: 1:250 Alexa Fluor 647-conjugated phalloidin (Invitrogen, A22287); 1:500 goat anti-mouse Alexa Fluor 488 (Invitrogen, A-11029). For the analyses, NMJs innervating muscle 4 on segments A2-A3 were imaged and analyzed for synaptic bouton quantification. Mature boutons are defined as boutons that are included in a chain of two or more boutons. Satellite boutons are defined as a single bouton that is not included in a chain of boutons, and instead, sprout off of a mature bouton or branch. The groups were compared using unpaired Student's t-test on GraphPad Prism software. p-Value less than 0.05 was considered statistically significant.

## *Drosophila* longevity assays

Longevity assay was modified from methods described previously (*Zhan et al., 2013*). For survival analysis, flies were aged at 27°C with no more than 15 flies per vial. Total more than 100 flies were used for each genotype. Vials were changed on a 2- to 3-day cycle. Death events were scored on a daily basis. Rescue in longevity was defined as greater than 5% increase in median lifespan in addition to the statistical threshold according to the Log-rank (Mantel-Cox) test, p<0.05. In the survival graphs shown, each set of experiments was done in the same time period with the corresponding control subjects in order to control longevity variation caused by environmental factors. Both genders were used in the survival assay unless otherwise specified.

## iPSC culture and motor neuron differentiation

A normal iPSC line (WC031i-5907-6, fibroblasts from neonatal male) was obtained from WiCell Research Institute (*Yin et al., 2019*). Into this line we introduced the following mutations using CRISPR/CAS9: P497H, 2XALS (P497H, P525S), 4XALS (P497H, P506T, P509S, P525S), and I498X, which harbors a 1 nt deletion in codon 497 that leads to frameshift and translation termination at codon 498. UBQLN2[P497H], UBQLN2[2XALS], and UBQLN2[4XALS] lines were sequenced for the top five ranking off-target cleavages (none were found) and confirmed for expression of pluripotency markers. All iPSCs have normal karyotypes, express stem cell markers, exhibit pluripotency (as assessed by capacity to differentiate into three germ layers), and were mycoplasma negative (*Yin et al., 2019*). STR analysis defines profile for each line, confirms clonality and purity to 95–98% confidence. Fifteen loci were tested, all matched appropriate source fibroblasts. iPSCs were cultured with mTeSR1 (Stemcell Technologies) on Matrigel

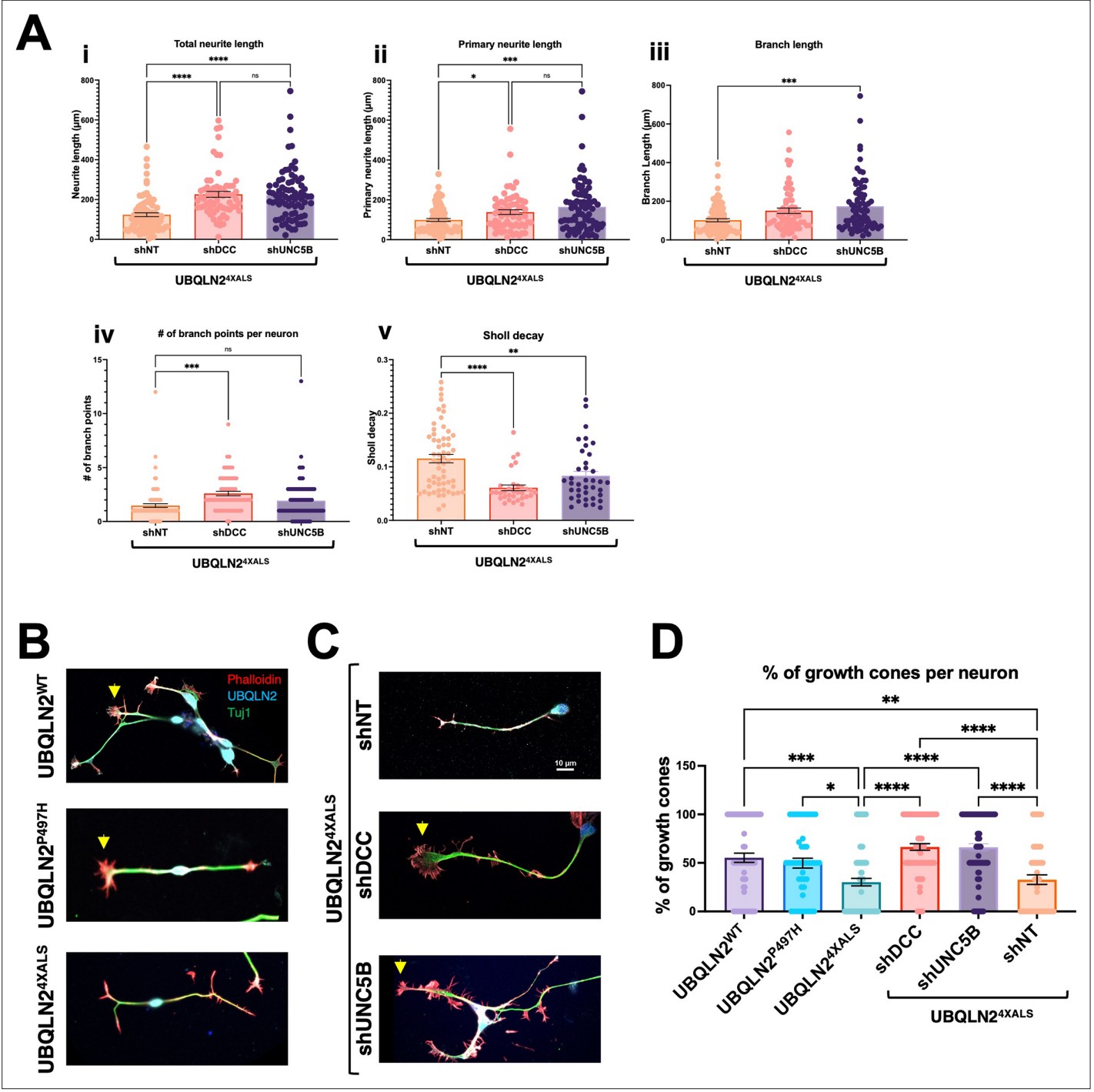

**Figure 9.** UNC5B and DCC silencing partially reverse neurite and growth cone defects in UBQLN2$^{4XALS}$ inducible motor neurons (iMNs). (**A**) UBQLN2$^{4XALS}$ iMNs expressing shNT, shUNC5B, or shDCC were subjected to Sholl analysis of neurite length and complexity as described in *Figure 7*. Data analysis was performed using ordinary one-way ANOVA. Data are shown as mean ± SEM. n>100 iMNs, *p≤0.05, **p≤0.01, ***p≤0.001, ****p≤0.0001. (**B**) Growth cone morphologies in UBQLN2$^{4XALS}$ iMNs. UBQLN2$^{WT}$, UBQLN2$^{P497H}$, and UBQLN2$^{4XALS}$ iMNs. iMNs of the indicated genotypes were stained with α-UBQLN2, α-Tuj1, and phalloidin. Note reduced growth cone elaboration in UBQLN2$^{4XALS}$ iMNs. (**C**) Enhanced growth cone elaboration in UBQLN2$^{4XALS}$ iMNs expressing DCC or UNC5B shRNAs. iMNs of the indicated genotype were stained with α-UBQLN2, α-Tuj1, and phalloidin. Arrowheads denote growth cones. (**D**) Quantification of growth cones in UBQLN2$^{WT}$, UBQLN2$^{P497H}$, and UBQLN2$^{4XALS}$, and UBQLN2$^{4XALS}$ iMNs expressing the indicated shRNAs. Data analysis was performed using ordinary one-way ANOVA. Data are shown as mean ± SEM. n>100 iMNs, *p≤0.05, **p≤0.01, ***p≤0.001, ****p≤0.0001.

The online version of this article includes the following source data and figure supplement(s) for figure 9:

*Figure 9 continued on next page*

*Figure 9 continued*

**Source data 1.** Data sets corresponding to *Figure 9A, D*.

**Figure supplement 1.** Expression of UNC5B and DCC mRNA in UBQLN2[4XALS] induced pluripotent stem cells (iPSCs) transduced with lentiviral shRNA vectors.

**Figure supplement 1—source data 1.** Data sets corresponding to *Figure 9—figure supplement 1A, B*.

**Figure supplement 2.** UNC5B or DCC knockdown does not affect UBQLN2[4XALS] aggregation in inducible motor neurons (iMNs).

(Corning). iPSCs on Matrigel were passaged with 0.5 mM EDTA. iPSC colonies were passaged every 4–7 days at a 1:3 to 1:6 split ratio.

Differentiation of iPSCs into iMNs was carried out as previously described (*Du et al., 2015*). In brief, iPSCs were dissociated and placed in Matrigel-coated plates. On the following day, the iPSC medium was replaced with a chemically defined neural differentiation medium, including DMEM/F12, Neurobasal medium at 1:1, 0.5×N2, 0.5×B27, and 1×Glutamax (all are from Invitrogen). CHIR99021 (3 μM, Torcris), 2 μM DMH-1 (Torcris), and 2 μM SB431542 (Stemgent) were added in the medium. The culture medium was changed every other day. Human iPSCs maintained under this condition for 7 days were induced into neuroepithelial progenitors (NEP). The NEP cells were then dissociated with dispase (1 mg/ml) and split at 1:6 with neural differentiation medium described above. Retinoic acid (RA, 0.1 μM, Stemgent) and 0.5 μM purmorphamine (Stemgent) were added in combination with 1 μM CHIR99021, 2 μM DMH-1, and 2 μM SB431542. The medium was changed every other day. NEP

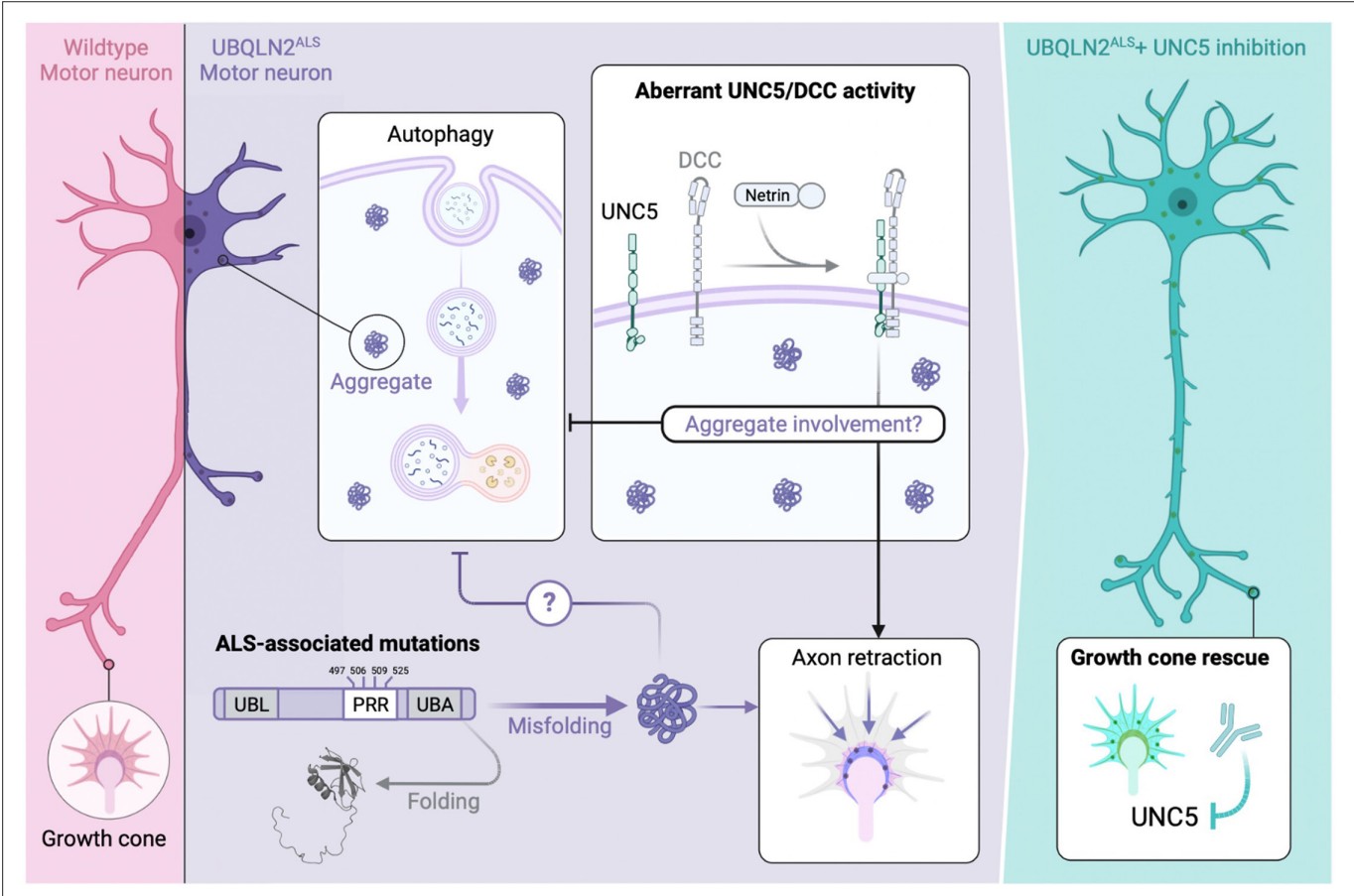

**Figure 10.** Speculative model for UBQLN2[ALS] toxicity suppression through the UNC5 pathway. (Left panel) Wild-type motor neurons exhibit healthy neurites and well-elaborated growth cones devoid of UBQLN2 aggregates. (Center panel) Amyotrophic lateral sclerosis (ALS)-associated mutations in the proline-rich repeat (PRR) of UBQLN2 promote its misfolding and assembly into aggregates that trigger UNC5/DCC-dependent axonal retraction. UBQLN2[ALS] aggregates may directly initiate pathologic UNC5/DCC signaling or impact other aspects of cellular regulation, including the endolysosomal-autophagosomal pathway, leading to proteostasis deregulation and growth cone retraction. (Right panel) Suppression of UNC5/DCC signaling with rationally designed antibodies or small molecules (not shown) rescues growth cone defects in UBQLN2[ALS] motor neurons.

cells maintained under this condition for 7 days differentiated into OLIG2+ motor neuron progenitors (MNPs). To induce motor neuron differentiation, OLIG2+ MNPs were dissociated with EDTA (0.5 mM) and cultured in suspension in the above neural differentiation medium with 0.1 μM RA and 0.1 μM purmorphamine. The medium was changed every other day. OLIG2+ MNPs under this condition for 6 days differentiated into HB9+ NMPs. The HB9+ NMPs were then dissociated with Accutase (Invitrogen) into single cells and plated on Matrigel-coated plates. The HB9+ NMPs were cultured with 0.1 μM RA, 0.1 μM purmorphamine, and 0.1 μM Compound E (Millipore) for 6 days to mature into CHAT+ iMNs.

## Microscopy

For immunostaining, iPSCs and iMNs were fixed with 4% paraformaldehyde in PBS, permeabilized with 0.2% PBST, blocked with 2% BSA, and stained with primary antibodies for overnight at 4°C and then stained with α-rabbit-Alex-488 and α-mouse-Alexa-594-conjugated secondary antibodies. Images were acquired using a Nikon A1 confocal microscope using either a 20× lens or a 60× oil lens. For Lysotracker assays, iPSCs were incubated with 70 nM LysoTracker Red DND-99 (L7528, Invitrogen) for 1 hr at 37°C in culture medium. Live microscopy of lysosomes was performed using a Nikon A1 confocal microscope with a heated chamber and an objective to maintain the cells at 37°C using a 60× oil lens.

## Immunoblotting

Fly lysates were prepared by homogenizing 10 fly heads in 50 μl of RIPA lysis buffer. The soluble fraction was taken after centrifugation at 21,000 × $g$ in a microcentrifuge for 10 min iPSC extract were prepared using in lysis buffer containing 20 mM Tris-HCl (pH 8.0), 138 mM NaCl, 10 mM KCl, 1 mM MgCl$_2$, 1 mM EDTA, and 1% Triton-X 100 vol/vol (TX buffer). All lysis buffers were supplemented with protease inhibitor cocktail (Sigma, P8340), 10 mM NaF, and 1 mM DTT. Following centrifugation at 21,000 × $g$ for 15 min, the insoluble pellet was washed twice with PBS, then suspended and boiled in Laemmli buffer. For immunoblotting, samples were separated by SDS-PAGE and transferred to PVDF membranes and immunoblotted with primary antibodies and LI-COR IRDye secondary antibodies (IRDye 800CW goat anti-rabbit and IRDye 680RD goat anti-mouse) as described. Signals were acquired using Odyssey bio-systems (LI-COR Biosciences). Immunoblotting results were analyzed and organized with ImageStudio Lite software (LI-COR).

## Limited proteolysis

iPSCs were lysed in buffer containing 20 mM Tris-HCl (pH 8.0), 138 mM NaCl, 10 mM KCl, 1 mM MgCl$_2$, 1 mM EDTA, and 0.2% NP-40 vol/vol. A total of 20 μg of each protein lysate in 10 μl was digested with increasing amounts of chymotrypsin (0.05–0. 2 μg/ml, final volume, Sigma) for 5 min at room temperature. Digestion was terminated by addition of Laemmli loading dye and boiling at 95°C for 5 min. The digested proteins were analyzed by Western blotting with anti-UBQLN2 antibodies.

## Sholl analysis

Sholl analysis was done by determining the number of neurite branches at various radial distances from the cell body. The rate at which branching decreases as a function of distance is the Sholl regression coefficient, or Sholl decay. Digitizing individual neurite branching patterns, or tracing, was performed using the Simple Neurite Tracer (SNT) plugin in ImageJ. Semi-automatic tracing of β-tubulin staining in iMNs was carried out blinded on individual neurons. SNT's Python application programming interface was implemented to measure the number of branch points and their respective radial distances. Other morphological descriptors including neurite length, branching number, and primary neurite number were measured in a similar way and graphed as scatter plots in GraphPad. Using the Python programming language, linear regression was fitted to the semi-log plots of the total branch points for every 1 μm radial distances.

There are a variety of ways to perform curve fitting for Sholl analysis, including linear mixed models. However, mixed models are necessary for samples with extensive clustering or heterogeneity (*Wilson et al., 2017*). Since each sample was differentiated, immunostained, and imaged simultaneously and in the same way, a simple linear regression model was appropriate. To ensure each linear fit was an accurate approximation of branch point distribution, only linear fits with a high Pearson's correlation

(R²>0.8) were used to calculate the Sholl coefficient (*Equation 1*). Individual neuron Sholl decay was graphed as scatter plots using GraphPad, where each neuron was one data point. Since neurons with a low Pearson's correlation were excluded in this analysis, all neurons of a given genotype were combined into one Sholl plot, in an additional, more inclusive analysis. This was accomplished by plotting the average number of branch points at 1 µm intervals among all neurons of a given genotype. Due to high variation of branching within each sample, only branch points that fell within the 10–90 percentile were linearly fitted to yield the overall Sholl decay for each genotype.

$$\log\left(\frac{N}{pr^2}\right) = -kr + m \tag{1}$$

$$N = \text{\# branch points (intersections)}$$
$$r = \text{radial distance from soma}$$
$$k = \text{Sholl regression coefficient}$$
$$m = \text{y intercept of linear fitted line}$$

## Growth cone analysis

Neurite terminals to the cell body were classified as growth cones (filopodial and lamellipodial ends, *Figure 9B*, arrows in UBQLN2$^{WT}$ and UBQLN2$^{P497H}$) or blunt ends (*Figure 9B* UBQLN2$^{4XALS}$). The percent of neurite terminals classified as growth cones was measured for individual neurons.

## Pearson's correlation coefficients for colocalization analysis

For the fluorescence quantification of colocalization images, neuronal cell body regions were extracted from each image using ImageJ/Fiji's selection tool. Thresholding of every region removed background signals. Pearson's colocalization coefficients of each cell body region were determined using ImageJ/Fiji's colocalization package. Pearson's coefficients were plotted as a bar graph to compare the colocalization (rho = 1) or exclusion (rho = –1) of UBQLN2 with LC3, Rab5, LAMP1, or P62. Ten sight fields for each group were analyzed.

## Statistical processing

Statistical analysis information including individual replicates and biological replicates number, mean or median, and error bars are explained in the figure legends. The statistical tests and resulting p-values are shown in the figure legends and/or figure panels.

## Acknowledgements

The authors would like to thank Lance A Rodenkirch (UW-Madison) for imaging assistance and Dr. Ludo Van Den Bosch, Dr. Katarina Ditlau, and Lisha Ye (University of Lueven) for helpful advice. This work was supported by grants from the National Institutes of Health (1RF1AG069483-01A1 to RST); National Institutes of Health/NCI (1R21 NS101661-01-A1 to RST); ALS Association (Proteostatic regulation by Ubiquilins in ALS to RST); National Institute on Aging (R21 AG065896-01A1 to SHK).

# Additional information

## Funding

| Funder | Grant reference number | Author |
| --- | --- | --- |
| National Institute on Aging | R21 AG065896-01A1 | Sang Hwa Kim |
| National Institutes of Health | 1RF1AG069483-01A1 | Randal S Tibbetts |
| National Cancer Institute | 1R21 NS101661-01-A1 | Randal S Tibbetts |
| ALS Association | Proteostatic regulation by Ubiquilins in ALS | Randal S Tibbetts |

| Funder | Grant reference number | Author |
|--------|------------------------|--------|

The funders had no role in study design, data collection and interpretation, or the decision to submit the work for publication.

## Author contributions

Sang Hwa Kim, Conceptualization, Resources, Data curation, Software, Formal analysis, Supervision, Funding acquisition, Validation, Investigation, Visualization, Methodology, Writing – original draft, Writing – review and editing; Kye D Nichols, Software, Formal analysis, Validation, Visualization; Eric N Anderson, Formal analysis, Validation, Visualization, Methodology, Writing – review and editing; Yining Liu, Nandini Ramesh, Formal analysis, Validation, Visualization; Weiyan Jia, Formal analysis, Visualization; Connor J Kuerbis, Validation, Visualization; Mark Scalf, Formal analysis, Validation, Methodology; Lloyd M Smith, Resources, Methodology; Udai Bhan Pandey, Data curation, Writing – review and editing; Randal S Tibbetts, Conceptualization, Resources, Data curation, Formal analysis, Supervision, Funding acquisition, Investigation, Methodology, Writing – original draft, Writing – review and editing

## Author ORCIDs

Sang Hwa Kim ⓘ http://orcid.org/0000-0002-9245-4514
Yining Liu ⓘ http://orcid.org/0000-0003-2632-7119
Lloyd M Smith ⓘ http://orcid.org/0000-0002-6652-8639
Udai Bhan Pandey ⓘ http://orcid.org/0000-0002-6267-0179
Randal S Tibbetts ⓘ http://orcid.org/0000-0003-2245-2297

## Decision letter and Author response

Decision letter https://doi.org/10.7554/eLife.84382.sa1
Author response https://doi.org/10.7554/eLife.84382.sa2

# Additional files

## Supplementary files

• MDAR checklist

## Data availability

The RNA sequencing raw dataset has been deposited to Dryad and is accessible at https://doi.org/10.5061/dryad.tdz08kq39. All data generated or analyzed during this study are included in the manuscript.

The following dataset was generated:

| Author(s) | Year | Dataset title | Dataset URL | Database and Identifier |
|-----------|------|---------------|-------------|--------------------------|
| Kim S, Tibbetts RS, Jia W, Liu Y | 2022 | Transcriptomic analysis of UBQLN2-ALS *Drosophila* | https://dx.doi.org/10.5061/dryad.tdz08kq39 | Dryad Digital Repository, 10.5061/dryad.tdz08kq39 |

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
