## [Editor Report]

This valuable study carried out a genetic screening of *Drosophila* lines expressing wild-type or ALS/FTD mutations of ubiquilin 2 and identified several suppressors and enhancers of ubiquilin 2 phenotypes. The study particularly focused on two genes involved in axon guidance pathways, unc5, and beat-1b. The evidence supporting the conclusions is solid and will be of interest to a broad audience studying ALS/FTD and neurodegenerative diseases.

---

## [Decision Letter]

**Decision letter after peer review:**

Thank you for submitting your article "Axon guidance genes modulate neurotoxicity of ALS-associated UBQLN2" for consideration by *eLife*. Your article has been reviewed by 2 peer reviewers, and the evaluation has been overseen by a Reviewing Editor and Jeannie Chin as the Senior Editor. The following individual involved in review of your submission has agreed to reveal their identity: Haining Zhu (Reviewer #1).

Essential revisions:

In addition to addressing the specific comments of the Reviewers (reviews appended below), please address these Essential Revisions below.

1) Much of the data presented is qualitative, without proper quantification and statistical analysis. All data need to be quantified and presented with appropriate statistics.

2) The heat shock effect in the *Drosophila* lines was not clear in the study. Why did some lines show phenotypes only at 29C but not 22C? Ubiquilin 2 expression was not impacted by 29C, then what caused the phenotypic differences? In the method sections please describe clearly whether a temperature sensitive promoter was used in the flies.

3) Data on male and female flies were shown separately in some but not all experiments. Please discuss whether there was a sex difference in those experiments.

4) Quite bit of data presented appear to be peripheral with no significant contribution to the main findings. Moreover, some data were introduced but were not explained. For instance, the RNA-Seq analysis (Figure 2) did not contribute much to the study. The rescue effect of UBA* (F594A mutant) in Figure 1-Supplemental 1B was interesting but was not elaborated or followed up. FUS flies in Figure 6-Supplement 2 were abruptly introduced with little discussion. Please remove data that is unrelated to the main findings of the paper. Some data can be moved to the Supplemental data, but only if they are integrate well with the story and are clearly described.

5) The main quadrupole (4xALS) mutation used in the study was not found in patients. The relevance of the findings needs to be thoroughly justified.

6) ALS and FTD are age-related neurodegenerative diseases, whereas the involvement of axon guidance genes in indicative of disruptions during the developmental stage. Please discuss this potential caveat.

7) If new lines were made for this paper, then proper characterization should be presented (staminality markers etc).

*Reviewer #1 (Recommendations for the authors):*

Specific comments on individual figures.

Figure 1:

(1) Can the eye phenotype be quantified? Some mutations appeared to cause relatively minor phenotypes.

(2) Any sex differences? Why so?

(3) No data were presented for I498X flies.

(4) Why did UBA* (F594A) rescued the phenotype?

(5) It appears that ubiquilin 2-UBA* was exclusively outside the nucleus in images shown in Supplement 1C. Why?

(6) "single copy" "two copies" used in the first paragraph of the Results section (page 5) are confusing. Did they refer to the number of copies of transgene in flies? Or did they refer to single, double, or quadrupole mutations? This needs to be clarified.

(7) Most importantly, what caused the heat shock effect (29C vs 22C) on the phenotypes? The study ruled out the effect of temperature on ubiquilin 2 expression, but did not provide any explanation for this interesting observation.

Figure 2: The data are loosely related and don't contribute much to the study.

Figure 4: Why shRab5 only showed effect in 4xALS flies at 29C but not 22C? Why did shRab5 showed effect at both 22C and 29C in WT and P497H flies?

Figure 5: What temperature was used for Figure 5B? 22C or 29C?

Figure 7:

(1) Any sex differences? Why so?

(2) D42-Gal4 or D42-WT ubiquilin 2 should be included as a control in Figure 7B-C.

(3) Supplement 3. Two blues curves are too similar. A different color is suggested so that they can be easily distinguished. In addition, Elev-shUnc5 appeared to have a marginal effect in females (panel B).

Figure 9: The western blot results need to be quantified to better understand the effects described in the text.

*Reviewer #2 (Recommendations for the authors):*

– All WB and eye phenotype need quantification whenever a comparison is made. All data should be accompanied by quantification graphs, showing statistical significances etc.

– Mass Spec in figure 1 needs stats.

– page 8 line 20: why should GFP-Rab5 rescue the eye phenotype of the UBQLN2wt and p497h which do not have eye phenotype at 29{degree sign} (according to Figure 1)?

– Figure suppl 1 C: why is here a ELAV promoter used? In the 4XALS, UBQLN2 is greatly aggregating. This I am sure will be reflected in a sol/insol RIPA extraction which has not been performed with this particular mutant under this particular promoter.

– Figure 2: I totally miss the relevance of this figure for the entire paper. That the flies are different is clear from the eye degeneration phenotype. When the reader comes to this figure, he/she is brought to try to find the connection with the rest of the paper which is hard to find. It would be more relevant for example to show if the suppressor and the enhancer found in the screening are different in this RNAseq data. Moreover, I think in Figure 2B that the reference group to compare all the other should be GMR and not WT. The background is for all GMR not WT so for which reason did the authors compare all the group to the WT and then build the Venn diagram? I think this is a mistake and might lead to misleading findings.

– Figures 4 and 5 are a nice validation of the screening, but since the findings are no further investigated these figures might be better put in the supplementary.

– Figure 6D: WB for Unc5 missing.

– Figure suppl 3 should be moved in the main figure 7.

– Figure 9: if these are new lines made for this paper a proper characterization should be presented (staminality markers etc). Besides this, the entire figure 9 should go in the supplementary as it functions only as a support for the data in the neurons. I also struggle to see the value of all the data with the BafA1 and of the lysosome analysis if they are no reproduced in neurons.

– Figure suppl 8 is missing the WB for DCC to show reduced level of the protein as well as of the RNA.

[Editors’ note: further revisions were suggested prior to acceptance, as described below.]

Thank you for resubmitting your work entitled "Axon guidance genes modulate neurotoxicity of ALS-associated UBQLN2" for further consideration by *eLife*. Your revised article has been evaluated by David Ron (Senior Editor) and a Reviewing Editor, as well as the original Reviewers.

The manuscript has been improved but there are some remaining issues that need to be addressed, as outlined below:

1. Please include the Eye Degeneration Score in all of the figures. It can perhaps be omitted in the screening to avoid making the figure too dense, but it will be helpful in the other figures to ensure that people outside of the fly field can evaluate the differences. Including the Eye Degeneration Score will also allow evaluation of the consistency of the findings across experiments.

2. Please ensure that the Images of the fly eyes have high enough resolution to be clear (for example, Figure 3 Supplementary 1 is clear).

3. Please include a quantification of Figure 8C: if a decreased co-localization with these markers is claimed, Pearson's coefficient or fluorescence intensity should be added.

4. Consider revising the choice of key words for this work to better reflect the content of the paper.

---

## [Author Response]

Essential revisions:In addition to addressing the specific comments of the Reviewers (reviews appended below), please address these Essential Revisions below.1) Much of the data presented is qualitative, without proper quantification and statistical analysis. All data need to be quantified and presented with appropriate statistics.

We have added multiple replicates, quantifications and additional statistical analyses for several figures in which we claimed a difference between experimental samples (Figure 1—figure supplement 1B, 1C, Figure 2—figure supplement 1E, 1H, Figure 2—figure supplement 2C, Figure 3E, Figure 5F, and Figure 6E). Western blots showing UBQLN2 fractionation in iPSCs (Figure 6A) were quantified but we did not statistically analyze the data because the amount of UBQLN2 in the insoluble fractions was consistently too low for confident analysis. However, we added the triplicate western blotting data to the Figure 6A source data. In those instances where insoluble UBQLN2 was detected, the band intensity was qualitatively the same across all experimental genotypes.

2) The heat shock effect in the *Drosophila* lines was not clear in the study. Why did some lines show phenotypes only at 29C but not 22C? Ubiquilin 2 expression was not impacted by 29C, then what caused the phenotypic differences? In the method sections please describe clearly whether a temperature sensitive promoter was used in the flies.

Figure 1—figure supplement 1C: The heat inducibility of the UBQLN2 transgenes, including UBQLN2^4XALS^ can be partly attributed to heat shock elements in the UAS promoter. This was noted in the text (page 6, line 4-14) and has been highlighted in the revised Materials and methods (page 25, line 6-8). The heat inducibility of dUbqln is interesting and may reflect transcriptional and/or posttranscriptional mechanisms. While it is possible that increased UBQLN2 contributes to the severe phenotypes in UBQLN2^4XALS^ flies reared at 29°C; this is not seen for UBQLN2^WT^ and UBQLN2^P497H^ flies. Instead, we postulate that heat stress synergizes with the misfolded UBQLN2^4XALS^ protein to disrupt proteostasis and endolysosomal function. We have added a point on this matter in paragraph 2 of the Discussion (page 16, line 15-25) section of the revised manuscript.

3) Data on male and female flies were shown separately in some but not all experiments. Please discuss whether there was a sex difference in those experiments.

UBQLN2^4XALS^-associated toxicity phenotypes were seen in both sexes across all phenotypic assays; however, subtle sex differences were observed in two experiments. In Figure 4D, Unc-5 silencing extended the lifespan of Elav>Gal4 female control flies but not Elav>Gal4 male control flies and, in Figure 4A, an Unc-5 KK RNAi line rescued climbing of D42>UBQLN2^4XALS^ male flies, but not female flies (a second Unc-5 RNAi line rescued both males and females). The sexual dimorphic phenotype in this experiment was noted in the results p10, line 20-21.

4) Quite bit of data presented appear to be peripheral with no significant contribution to the main findings. Moreover, some data were introduced but were not explained. For instance, the RNA-Seq analysis (Figure 2) did not contribute much to the study. The rescue effect of UBA* (F594A mutant) in Figure 1-Supplemental 1B was interesting but was not elaborated or followed up. FUS flies in Figure 6-Supplement 2 were abruptly introduced with little discussion. Please remove data that is unrelated to the main findings of the paper. Some data can be moved to the Supplemental data, but only if they are integrate well with the story and are clearly described.

We understand the reviewer’s point or the reviewer’s point is well taken. Appreciating the reviewer’s comment, we moved both figures to the supplementary data.

RNA-Seq (Figure 1—figure supplement 2)

Although not essential, the RNA-Seq adds experimental rigor to the study by providing strong molecular correlates to eye degeneration phenotypes across different UBQLN2 genotypes. It shows the unique toxicity of UBQLN2^4XALS^ and reinforces phenotypic similarity between UBQLN2^WT^ and UBQLN2^P497H^ flies, which likely reflects non-specific toxicity of overexpressed UBQLN2 proteins. We have carried out additional data analyses requested by the reviewer and moved the RNA-Seq data to Figure 1—figure supplement 2.

Lack of genetic interaction between FUS and Unc-5 (Figure 3—figure supplement 1).

This data was included to show that shUnc-5 is not a general suppressor of eye toxicity in *Drosophila*. This contrasts with *lilliputian*, whose mutation rescues toxicity phenotypes elicited by FUS, TDP-43, and UBQLN2. We believe that the FUS control data enhances experimental rigor and have retained the data in the revised manuscript, with some additional clarification on page 10, line 3-7.

UBA mutant (Figure1—figure supplement 1).

Both aggregation and toxicity of UBQLN2^4XALS^ were abolished by a F594A mutation in the UBA domain that abolishes Ub binding. However, as the reviewer noted, we did not follow up on the finding and have therefore chosen to remove it from the revised manuscript.

5) The main quadrupole (4xALS) mutation used in the study was not found in patients. The relevance of the findings needs to be thoroughly justified.

The use of combinatorial mutants—either in the same gene or same pathway—can sometimes be used to enhance neurodegenerative phenotypes in cellular and rodent models for neurodegenerative diseases, most notably, Alzheimer’s Disease. In the case of the 4XALS mutant, we reasoned that its enhanced aggregation might drive stronger phenotypes than those elicited by UBQLN2 clinical alleles, whose toxicity is marginally discernible from overexpressed UBQLN2^WT^. We have clarified the rationale for testing the 4XALS mutant and articulated its potential strengths and weaknesses in Results (page 5, line 14-page 6, line 2) and Discussion (page 16, line 15-25) sections.

6) ALS and FTD are age-related neurodegenerative diseases, whereas the involvement of axon guidance genes in indicative of disruptions during the developmental stage. Please discuss this potential caveat.

We have inserted the following sentence in the discussion to note this caveat:

“Consistent with this notion, *UNC5B* has been linked to neurodegeneration in the 6-OHDA model of Parkinson’s Disease (PD) and *UNC5C* has been nominated as a risk allele in late-onset Alzheimer’s Disease. Defining the contributions of pathologic UNC5 signaling to the development or progression of ALS-dementia awaits further study.” on Page 20, line 2-6.

We have added a similar sentence to the Limitations paragraph at the end of the Discussion:

“Third, it is possible that axon guidance genes are only relevant to UBQLN2 toxicity in the context of the developing nervous system”.

7) If new lines were made for this paper, then proper characterization should be presented (staminality markers etc).

All lines are derived from a male iPSC line that has been karyotyped and characterized for stemness and differentiation potential (Yin et al. (2019) Stem Cell Res 34, 101365.)(1). We also karyotyped the P497H, 2XALS, and 4XALS Clone 1 line used throughout the paper.

Reviewer #1 (Recommendations for the authors):Specific comments on individual figures.Figure 1:(1) Can the eye phenotype be quantified? Some mutations appeared to cause relatively minor phenotypes.

For the genetic screen we scored the rough eye phenotype according to the rubric shown in Figure 2B. For subsequent eye morphology studies, we simply relied on qualitative differences between UBQLN2^4XALS^ flies in the absence or presence of different test mutations. While this type of qualitative analyses is standard in the field and generally worked well for our studies, it lacks the power to distinguish subtle phenotypic differences between different mutants and we generally did not follow up on mutants that showed marginal effects. As a consequence, our screen likely missed weak modifiers of the UBQLN2^4XALS^ phenotype. To reduce confounding effects of genetic background differences we performed a side-by-side comparison of F1 progeny harboring the UBQLN2^ALS^ transgene and test mutation/chromosome or the UBQLN2^ALS^ transgene and control (e.g. balancer) chromosome.

(2) Any sex differences? Why so?

UBQLN2^4XALS^ was toxic in both male and female flies under all experimental conditions and UBQLN2 modifier genes yielded similar phenotypes in both male and female flies with a couple of exceptions. For example, Unc-5 silencing extended the lifespan of control Elav>Gal4 male flies, but not control female Elav>Gal4 flies (Figure 4D), and a validating Unc-5 RNAi line (KK) strongly rescued the climbing phenotype of male, but not female D42>UBQLN2^4XALS^ flies (Figure 4A). The reasons for these differences are not clear.

(3) No data were presented for I498X flies.

We did not generate I498X flies, only the I498X iPSC line which was a byproduct of UBQLN2 CRISPR mutagenesis.

(4) Why did UBA* (F594A) rescued the phenotype?

The simplest explanation is that UBQLN2^4XALS^ must bind to Ub—possibly in the form of ubiquitylated substrates—to aggregate in cells. Another (less interesting) explanation is that the UBA* mutation destabilizes UBQLN2^4XALS^ such that it does not accumulate to levels required for its aggregation. Regardless, we agree that the UBA data is peripheral to the main story and removed Figure 1—figure supplement 1B, and 1C from the revised manuscript.

(5) It appears that ubiquilin 2-UBA* was exclusively outside the nucleus in images shown in Supplement 1C. Why?

This is an interesting point that we had not noted previously. We may revisit this result in future studies.

(6) "single copy" "two copies" used in the first paragraph of the Results section (page 5) are confusing. Did they refer to the number of copies of transgene in flies? Or did they refer to single, double, or quadrupole mutations? This needs to be clarified.

We apologize for the lack of clarity. We generated flies harboring single-copy UAS-UBQLN2 transgenes on Chr2 and Chr3 that were then recombined with different Gal4 driver lines. We have replaced the confusing language on page 5, line 14-page 6 line 2 with the more appropriate terms “hemizygous” and “homozygous” to describe flies harboring one or two UBQLN2 alleles. We have also elaborated on the construction of the flies in the Materials and methods section.

(7) Most importantly, what caused the heat shock effect (29C vs 22C) on the phenotypes? The study ruled out the effect of temperature on ubiquilin 2 expression, but did not provide any explanation for this interesting observation.

The most plausible hypothesis is that heat stress synergizes with the misfolded UBQLN2^4XALS^ protein to disrupt proteostasis and/or endolysosomal function. The following clarifying language has been added to paragraph 2 of the Discussion (page 16, line 15-25): “The reason for enhanced toxicity of UBQLN2^4XALS^ is unclear; however, its enhanced aggregation potential may overwhelm cellular proteostasis machinery and/or accelerate disease mechanisms that are slow to manifest in neurons harboring ALS point mutations. This is consistent with the fact that UBQLN2^4XALS^ toxicity in flies was unmasked by HS, which is a well-known inducer of proteotoxicity.” We have also explicitly state the HS inducibility of the UAS-Gal4 in the Materials and methods (page 25, line 6-8).

Figure 2: The data are loosely related and don't contribute much to the study.

Although not essential, the RNA-Seq adds experimental rigor to the study by providing strong molecular correlates to eye degeneration phenotypes across different UBQLN2 genotypes. It shows the unique toxicity of UBQLN2^4XALS^ and reinforces phenotypic similarity between UBQLN2^WT^ and UBQLN2^P497H^ flies, which likely reflects non-specific toxicity of overexpressed UBQLN2 proteins. We have carried out additional data analyses requested by the reviewer (Figure 1—figure supplement 2A) and moved the RNA-Seq data to Figure 1—figure supplement 2.

Figure 4: Why shRab5 only showed effect in 4xALS flies at 29C but not 22C? Why did shRab5 showed effect at both 22C and 29C in WT and P497H flies?

Although there are several possibilities, we speculate that the enhanced solubility of UBQLN2^WT^ and UBQLN2^P497H^ (relative to UBQLN2^4XALS^) leads to a greater disruption of the ubiquitin and endolysosomal pathways at room temperature (as reported Kim et al., Human Molecular Genetics, 2018, Vol. 27, No. 2 p322–337)(2). As a consequence, UBQLN2^WT^ and UBQLN2^P497H^ lines are more sensitive to reductions in Rab5 gene dosage than UBQLN2^4XALS^ flies at room temperature. We have attempted to clarify this in the Results (page 8, line 21-26) and Discussion (page 19, line 4-7) of the revised manuscript.

Figure 5: What temperature was used for Figure 5B? 22C or 29C?

The experiment was carried out at 29°C. We added specific details to the figure legend.

Figure 7:(2) D42-Gal4 or D42-WT ubiquilin 2 should be included as a control in Figure 7B-C.

We added data for the D42-Gal4 control in Figure 4B and 4C.

(3) Supplement 3. Two blues curves are too similar. A different color is suggested so that they can be easily distinguished. In addition, Elev-shUnc5 appeared to have a marginal effect in females (panel B).

We made the suggested color change. shUnc-5 silencing increased lifespan of males and female flies (now presented in Figure 4D).

Figure 9: The western blot results need to be quantified to better understand the effects described in the text.

We have added multiple replicates, quantifications and additional statistical analyses for several figures in which we claimed a difference between experimental samples (Figure 1—figure supplement 1B, 1C, Figure 2—figure supplement 1E, 1H, Figure 2—figure supplement 2C, Figure 3E, Figure 5F, and Figure 6E). Western blots showing UBQLN2 fractionation in iPSCs (Figure 6A) were quantified but we did not statistically analyze the data because the amount of UBQLN2 in the insoluble fractions was consistently too low for confident analysis. However, we added the triplicate western blotting data to the Figure 6A source data. In those instances where insoluble UBQLN2 was detected, the band intensity was qualitatively the same across all experimental genotypes.

Reviewer #2 (Recommendations for the authors):– All WB and eye phenotype need quantification whenever a comparison is made. All data should be accompanied by quantification graphs, showing statistical significances etc.

We have added multiple replicates, quantifications and additional statistical analyses for several figures in which we claimed a difference between experimental samples (Figure 1—figure supplement 1B, 1C, Figure 2—figure supplement 1E, 1H, Figure 2—figure supplement 2C, Figure 3E, Figure 5F, and Figure 6E). Western blots showing UBQLN2 fractionation in iPSCs (Figure 6A) were quantified but we did not statistically analyze the data because the amount of UBQLN2 in the insoluble fractions was consistently too low for confident analysis. However, we added the triplicate western blotting data to the Figure 6A source data. In those instances where insoluble UBQLN2 was detected, the band intensity was qualitatively the same across all experimental genotypes.

– Mass Spec in figure 1 needs stats.

This has been added.

– page 8 line 20: why should GFP-Rab5 rescue the eye phenotype of the UBQLN2wt and p497h which do not have eye phenotype at 29{degree sign} (according to Figure 1)?

Although there are several possibilities, we speculate that the enhanced solubility of UBQLN2^WT^ and UBQLN2^P497H^ (relative to UBQLN2^4XALS^) leads to a greater disruption of the ubiquitin and endolysosomal pathways at room temperature (as reported Kim et al., Human Molecular Genetics, 2018, Vol. 27, No. 2 p322–337)(2). As a consequence, UBQLN2^WT^ and UBQLN2^P497H^ lines are more sensitive to reductions in Rab5 gene dosage than UBQLN2^4XALS^ flies at room temperature. We have attempted to clarify this in the Results (page 8, line 21-26) and Discussion (page 19, line 4-7) of the revised manuscript.

– Figure suppl 1 C: why is here a ELAV promoter used? In the 4XALS, UBQLN2 is greatly aggregating. This I am sure will be reflected in a sol/insol RIPA extraction which has not been performed with this particular mutant under this particular promoter.

Elav was used to examine UBQLN2 aggregation in the brain. Although not shown in this paper, we previously published the RIPA insolubility of UBQLN2^4XALS^ expressed in fly brains under control of Elav (Kim et al., Human Molecular Genetics, 2018, Vol. 27, No. 2 p322–337 Figure 4A) (2). Figure 1—figure supplement 1A, B has been modified to focus on the expression of wild-type and mutant UBQLN2 proteins expressed under control of GMR at 22°C and 29°C.

– Figure 2: I totally miss the relevance of this figure for the entire paper. That the flies are different is clear from the eye degeneration phenotype. When the reader comes to this figure, he/she is brought to try to find the connection with the rest of the paper which is hard to find. It would be more relevant for example to show if the suppressor and the enhancer found in the screening are different in this RNAseq data. Moreover, I think in Figure 2B that the reference group to compare all the other should be GMR and not WT. The background is for all GMR not WT so for which reason did the authors compare all the group to the WT and then build the Venn diagram? I think this is a mistake and might lead to misleading findings.

While we could carry out additional RNA-seq experiments to correlate phenotypic rescue with gene expression changes in UBQLN2^4XALS^ flies in the absence or presence of Unc-5 LOF alleles, we don’t feel the experiment would add much to the paper at this juncture. We used UBQLN2^WT^ flies as the reference group because we had already established that wild-type UBQLN2 overexpression causes mild phenotypic abnormalities and Ub pathway deregulation in flies (Kim et al. 2018)(2). We have now performed the requested comparison to GMR in Figure1—figure supplement 2A. Consistent with UBQLN2 overexpression toxicity, all three experimental genotypes (WT, P497H, 4XALS) exhibited hundreds of gene expression differences relative to GMR-Gal4 flies.

– Figures 4 and 5 are a nice validation of the screening, but since the findings are no further investigated these figures might be better put in the supplementary.

We moved Figure 4 and 5 to Figure 2—figure supplement 1 and 2.

– Figure 6D: WB for Unc5 missing.

Unfortunately, anti-*Drosophila* Unc5 antibodies are not commercially available. We also attempted to generate an anti-*Drosophila* Unc5 mAb, but this antibody did not work. Unc-5 RT-qPCR was not sensitive enough to detect decrease in eye specific knock down from whole head total RNA.

– Figure suppl 3 should be moved in the main figure 7.

We moved Figure supplement 3 to Figure 4D.

– Figure 9: if these are new lines made for this paper a proper characterization should be presented (staminality markers etc). Besides this, the entire figure 9 should go in the supplementary as it functions only as a support for the data in the neurons.

All lines are derived from a male iPSC line that has been fully characterized for stemness and differentiation potential (Yin et al. (2019) Stem Cell Res 34, 101365.)(1). We also karyotyped the UBQLN2-P497H, 2XALS and 4XALS clone 1 and validated all findings from this clone with UBQLN2-4XALS clone 2. Because we have now complemented the iPSC/BafA1 findings with iMN data (Figure 8) we have chosen to keep the iPSC findings in the main figure panel (Figure 6 in the revised manuscript).

I also struggle to see the value of all the data with the BafA1 and of the lysosome analysis if they are no reproduced in neurons.

We carried out the suggested experiments in iMNs, where BafA1 strongly induced aggregation of UBQLN2^4XALS^, but not UBQLN2^WT^ or UBQLN2^P497H^. UBQLN2^4XALS^ aggregates were coincident with p62 aggresomes and were partially colocalized with LAMP1+ lysosomes. By contrast, UBQLN2 was only weakly colocalized with Rab5 or LC3 in BafA1-treated iMNs. This data corroborates and extends the iPSC data and is presented in Figure 8 of the revised manuscript.

– Figure suppl 8 is missing the WB for DCC to show reduced level of the protein as well as of the RNA.

We agree that it is preferable to corroborate the DCC knockdown result by WB. Unfortunately, we were unable to consistently detect DCC in iMN extracts using several well-characterized antibodies. We are confident that DCC levels were reduced based on RT-qPCR findings show in Figure 9—figure supplement 1.

1. Yin, Y., Petersen, A. J., Soref, C., Richards, W. D., Ludwig, T., Taapken, S., Berndt, E., Zhang, S. C., and Bhattacharyya, A. (2019) Generation of seven induced pluripotent stem cell lines from neonates of different ethnic backgrounds. Stem Cell Res 34, 101365

2. Kim, S. H., Stiles, S. G., Feichtmeier, J. M., Ramesh, N., Zhan, L., Scalf, M. A., Smith, L. M., Pandey, U. B., and Tibbetts, R. S. (2018) Mutation-dependent aggregation and toxicity in a *Drosophila* model for UBQLN2-associated ALS. Hum Mol Genet 27, 322-337

[Editors’ note: further revisions were suggested prior to acceptance, as described below.]

1. Please include the Eye Degeneration Score in all of the figures. It can perhaps be omitted in the screening to avoid making the figure too dense, but it will be helpful in the other figures to ensure that people outside of the fly field can evaluate the differences. Including the Eye Degeneration Score will also allow evaluation of the consistency of the findings across experiments.

The Eye Degeneration Scores were included below the fly eye images, following the scoring criteria outlined in Figure 2B.

2. Please ensure that the Images of the fly eyes have high enough resolution to be clear (for example, Figure 3 Supplementary 1 is clear).

Having gone through all the figures we agree that images taken with our laboratory stereoscope do not match the clarity of images generated on Dr. Pandey’s stereoscope shown in Figure 3—figure supplement 1. However, the eye depigmentation and necrosis features used for the side-by-side phenotypic comparisons can be easily discerned regardless of the stereoscope used. That said, we have replaced certain images with better quality images that were taken from the same fly cohorts at the same time. Panel replacements include: Figure 2C (C144), Figure 2 Supplementary 2B (lilli^7-2^), Figure 3B (ED2426, BSC346), Figure 3C (CyO, Unc-5^3^, Unc-5^8^, shUnc-5 (TRiP)), Figure 3F (fra (#31664)), and Figure 5D (shBeat-1b, Beat-1b (p-insertion)). In Figure 2—figure supplement 2, we eliminated the fly eye images of BSC180, ED4651, and C144 as they were redundant with those shown in Figure 2C. The clarity of *Drosophila* eye images was also enhanced in some cases by adjusting their sharpness and brightness.

3. Please include a quantification of Figure 8C: if a decreased co-localization with these markers is claimed, Pearson's coefficient or fluorescence intensity should be added.

Pearson's coefficient was incorporated into Figure 8D.

4. Consider revising the choice of key words for this work to better reflect the content of the paper.

We substituted kye words with amyotrophic lateral sclerosis, UBQLN2, protein aggregation, induced motor neuron, genetic screen, axon guidance. Please advise if these remain inadequate.